# Rapid changes in morphogen concentration control self-organized patterning in human embryonic stem cells

Idse Heemskerk[1†], Kari Burt[1], Matthew Miller[1], Sapna Chhabra[2],
M Cecilia Guerra[1], Lizhong Liu[1], Aryeh Warmflash[1,3]*

[1]Department of Biosciences, Rice University, Houston, United States; [2]Systems, Synthetic and Physical Biology Program, Rice University, Houston, United States; [3]Department of Bioengineering, Rice University, Houston, United States

**Abstract** During embryonic development, diffusible signaling molecules called morphogens are thought to determine cell fates in a concentration-dependent way. Yet, in mammalian embryos, concentrations change rapidly compared to the time for making cell fate decisions. Here, we use human embryonic stem cells (hESCs) to address how changing morphogen levels influence differentiation, focusing on how BMP4 and Nodal signaling govern the cell-fate decisions associated with gastrulation. We show that BMP4 response is concentration dependent, but that expression of many Nodal targets depends on rate of concentration change. Moreover, in a self-organized stem cell model for human gastrulation, expression of these genes follows rapid changes in endogenous Nodal signaling. Our study shows a striking contrast between the specific ways ligand dynamics are interpreted by two closely related signaling pathways, highlighting both the subtlety and importance of morphogen dynamics for understanding mammalian embryogenesis and designing optimized protocols for directed stem cell differentiation.
**Editorial note:** This article has been through an editorial process in which the authors decide how to respond to the issues raised during peer review. The Reviewing Editor's assessment is that all the issues have been addressed (see decision letter).
DOI: https://doi.org/10.7554/eLife.40526.001

*For correspondence:
Aryeh.Warmflash@rice.edu

Present address: †Department of Cell and Developmental Biology, University of Michigan MedicalSchool, Ann Arbor, United States

Competing interests: The authors declare that no competing interests exist.

## Introduction

Mammalian development depends crucially on diffusible signaling molecules called morphogens, that are thought to determine cell fates in a concentration-dependent manner (*Green et al., 1992*; *Wilson et al., 1997*; *Wolpert, 1969*), and protocols for directed stem cell differentiation are based on this picture (*Chambers et al., 2009*; *Loh et al., 2016*; *McLean et al., 2007*; *Mendjan et al., 2014*). However, in the vertebrate embryo, expression patterns of these morphogens change rapidly, simultaneous with large-scale cell movements, and therefore individual cells experience substantial changes in morphogen levels during differentiation (*Arnold and Robertson, 2009*; *Balaskas et al., 2012*; *Dessaud et al., 2007*; *Dubrulle et al., 2015*; *Kinder et al., 2001*; *van Boxtel et al., 2015*). Duration of ligand exposure must logically be a relevant parameter, as was confirmed in a number of contexts, including Activin/Nodal signaling in early Zebrafish development and Sonic Hedgehog (Shh) signaling in the mouse neural tube (*Dessaud et al., 2007*; *Sako et al., 2016*). However, duration is only one of a large number of features of a dynamic signal. It is unknown whether the precise time course of ligand exposure plays a role in cell fate decisions, and if so, whether different pathways interpret signaling histories differently. In analogy with human speech, which enables sophisticated communication by relying on temporal modulation of a single mode of signaling (sound), it is possible that complex information is encoded in developmental signals by temporal

modulation to enable a range of different responses to a single pathway. In addition to ligand concentration and duration ('integral'), cells may also be sensitive to ligand rates of change ('derivative'), and it has been suggested that adaptive signaling pathways allow cells to perform this derivative computation (*Sorre et al., 2014*; *Yi et al., 2000*).

These concepts have begun to be explored in mammalian cell culture systems. For the Nκfb pathway, an intricate relation between ligand dynamics, signaling response and target gene activation was found (*Hoffmann et al., 2002*; *Kellogg et al., 2017*; *Selimkhanov et al., 2014*). Further demonstrating the importance of changing ligand concentrations, a class of ERK target genes was recently shown to be activated more efficiently by pulses than sustained signaling in 3T3 cells (*Wilson et al., 2017*), and we have shown that the response to TGFβ in C2C12 mouse myoblasts reflects ligand rate of increase (*Sorre et al., 2014*; *Warmflash et al., 2012*). However, these ideas have not been applied to mammalian development or differentiation of pluripotent stem cells, and their relevance to developmental patterning remains unexplored.

Gastrulation is the first differentiation event of the embryo proper, when the germ layers are formed and the body axes are established. Nodal and BMP4 are morphogens crucial for gastrulation in vertebrates (*Winnier et al., 1995*). In the early mammalian embryo, BMP4 is required for both initiating gastrulation and specifying the dorsal-ventral axis, while Nodal maintains the pluripotent epiblast and is subsequently required for mesoderm and endoderm differentiation (*Conlon et al., 1994*). Each pathway has distinct receptor complexes that phosphorylate specific signal transducers, known as receptor-Smads, which then complex with the shared cofactor Smad4 and translocate to the nucleus to activate target genes (*Figure 1a*) (*Zinski et al., 2018*). Both Nodal and BMP4 have been claimed to act in a concentration-dependent manner based on classic Xenopus experiments (*Green et al., 1992*; *Gurdon et al., 1994*; *Wilson et al., 1997*). However, those experiments allow alternative interpretations (*Heemskerk and Warmflash, 2016*), and the role of BMP4 and Nodal ligand dynamics has not been investigated.

Micropatterned human embryonic stem cells (hESCs) self-organize into reproducible spatial domains corresponding to each of the germ layers and were recently established as a method to recapitulate human gastrulation in vitro (*Warmflash et al., 2014*). This system can be easily manipulated and observed. Further, in contrast to the micropatterned colonies, when hESCs are grown more sparsely, their response to exogenous signals is uniform and not dependent on secondary signals, allowing for dissection of the dynamics of response (*Nemashkalo et al., 2017*). Thus, the response of cells to dynamic signals can be systematically investigated in sparse culture, and this information can be used to unravel the complexity of self-organized pattern formation in micropatterned colonies. In this study, we take this approach and use hESCs to evaluate the role of changing concentrations of BMP4 and Activin/Nodal in cell-fate decisions associated with gastrulation. Unexpectedly, we find an important role for rapid concentration changes in Nodal pathway response, while the BMP pathway responds to concentration and duration of ligand exposure more directly.

## Results

### SMAD4 signaling response of hESCs to BMP4 is sustained but to Activin is adaptive

To investigate how sudden increases in BMP4 and Nodal levels are interpreted by hESCs, we performed live imaging of hESCs with GFP:SMAD4 in the endogenous locus (*Nemashkalo et al., 2017*), and quantified signaling strength as the ratio of nuclear and cytoplasmic SMAD4 intensity (*Figure 1—figure supplement 1a*). We found that a sudden increase in BMP4 leads to sustained SMAD4 signaling (*Figure 1b,d*), consistent with our previous work (*Nemashkalo et al., 2017*). In contrast, the response to addition of Nodal and its substitute Activin is strongly adaptive, that is transient, and returns to a signaling baseline of around 20% of the response peak (*Figure 1c,d*, *Figure 1—figure supplement 1i, j*), similar to the previously observed response to TGFβ in C2C12 cells (*Sorre et al., 2014*; *Warmflash et al., 2012*). The response to recombinant Nodal was weak at all doses, likely reflecting the low quality of recombinant Nodal. To elicit a response in Nodal required 2 μg/ml of recombinant protein, whereas the response to Activin was saturated by 5 ng/ml (*Figure 1f*, *Figure 1—figure supplement 1i, j*). However, when the responses to Nodal and Activin were normalized to their respective maxima, their dynamics were identical (*Figure 1—figure supplement 1j*). Given our focus on the

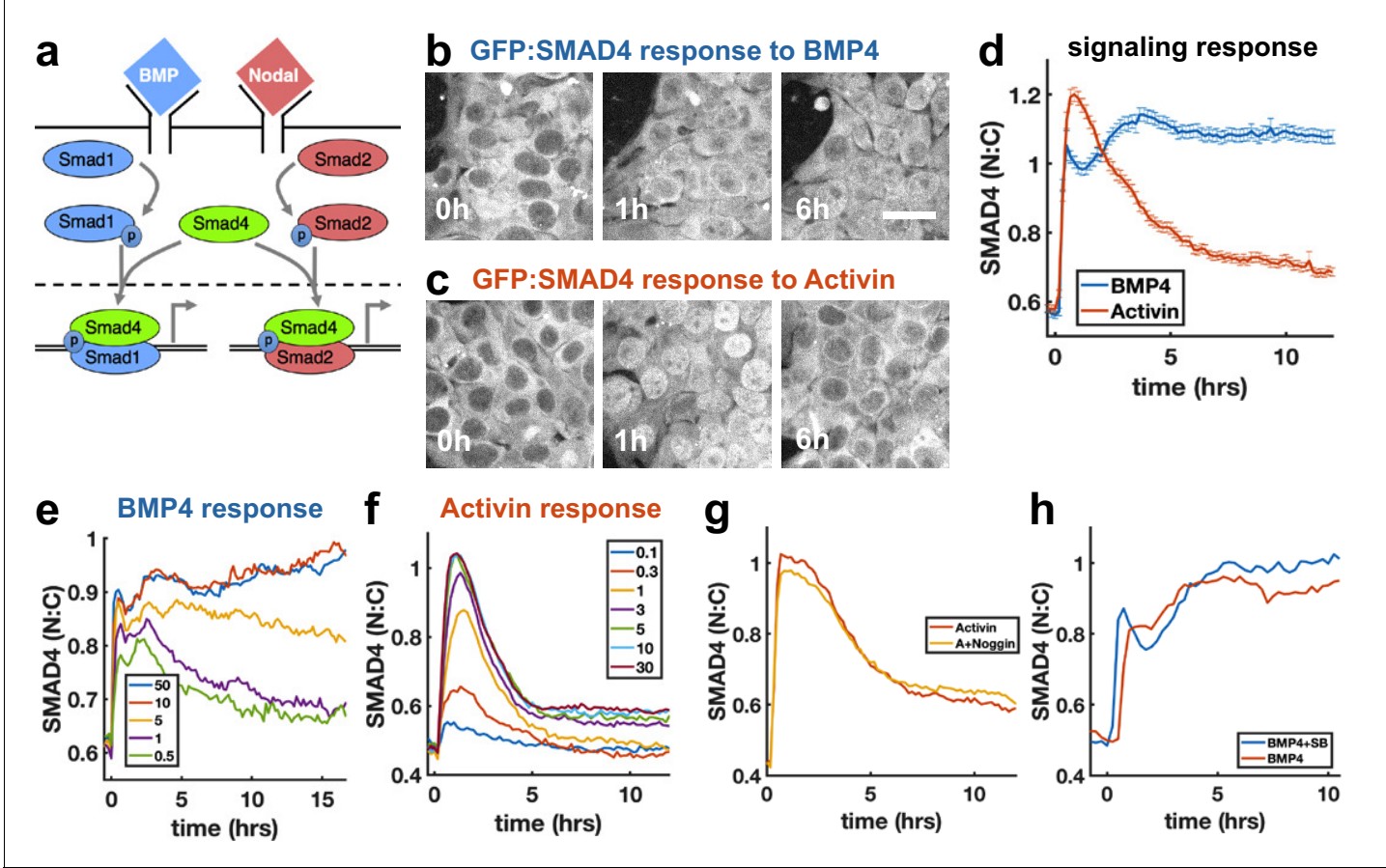

**Figure 1.** SMAD4 signaling response of hESCs to BMP4 is sustained while that to Activin is adaptive. (**a**) BMP and Nodal pathways share the signal transducer Smad4. (**b, c**) hESCs expressing GFP:SMAD4 at 0, 1 and 6 hr after treatment with BMP4 (**b**) or Activin (**c**). Scalebar 30 μm (**d**) GFP:SMAD4 average nuclear:cytoplasmic intensity ratio after treatment with BMP4 (blue) or Activin (red). Error bars represent standard error. Ncells ~ 700, distributions shown in (*Figure 1—figure supplement 1b–c*). (**e**) SMAD4 response to different doses of BMP4 shows decline at low doses with a dose-dependent time scale, suggesting ligand depletion. Doses in graph legend are in ng/ml. (**f**) SMAD4 signaling response to different doses of Activin shows fixed time scale of adaptation. (**g**) Quantification of GFP:SMAD4 nuclear to cytoplasmic ratio in response to either Activin alone or together with the BMP inhibitor Noggin (**h**) Quantification of GFP:SMAD4 nuclear to cytoplasmic ratio in response to either BMP alone or together with the Activin/Nodal inhibitor SB431542.

DOI: https://doi.org/10.7554/eLife.40526.002

The following video, source data, and figure supplement are available for figure 1:

**Source data 1.** MATLAB script and .mat files to reproduce the data panels in *Figure 1*.
DOI: https://doi.org/10.7554/eLife.40526.006

**Figure supplement 1.** Further characterization of response of hESCs to BMP4 and Activin.
DOI: https://doi.org/10.7554/eLife.40526.003

**Figure 1—video 1.** SMAD4 signaling response to BMP4 treatment is sustained.
DOI: https://doi.org/10.7554/eLife.40526.004

**Figure 1—video 2.** SMAD4 signaling response to Activin treatment is adaptive.
DOI: https://doi.org/10.7554/eLife.40526.005

dynamics of signaling, the extremely similar dynamics in response to Nodal and Activin, and the impracticality of performing experiments with Nodal given the concentrations required, we used Activin in all further experiments. The peak and baseline signaling levels, but not the timescale of adaptation, depended on the Activin dose (*Figure 1f*). For BMP4, there was a sharp response so that even low doses gave a nearly full initial response, however, the dose affected the duration of signaling in a way consistent with ligand depletion at low doses (*Figure 1e*, Appendix 1). Immunofluorescence staining for receptor-Smads revealed that SMAD1/5/8 activation mirrors the SMAD4 response to BMP4,

while in response to Activin, SMAD2/3 nuclear localization adapts less than SMAD4 to about 60% of peak response (*Figure 1—figure supplement 1e,g,h*). In each case, the response was due to the added ligand and not to induced secondary signaling through the other pathway, as addition of the BMP inhibitor Noggin had no effect on the dynamics in response to Activin, while addition of the Activin inhibitor SB431542 slightly increased the response to BMP (*Figure 1g,h*). Thus, endogenous BMP has no effect on the Nodal response, while Nodal signaling has a repressive effect on BMP. We note that some target genes may respond to SMAD2/3 without SMAD4 (*Levy and Hill, 2005*), and that since the dynamics of SMAD2/3 and SMAD4 are different, care is required in interpreting the dynamics of individual genes. Hereafter, when we refer to Nodal signaling as reflected by the SMAD4 reporter, we mean 'SMAD4-dependent Nodal signaling'.

## Adaptive signaling is caused by negative feedback controlling sequestration

Although TGFβ signaling through SMAD4 has been shown to adapt in C2C12 cells (*Sorre et al., 2014*; *Warmflash et al., 2014*), the molecular mechanisms remain unclear. A previous attempt to uncover relevant genes through a genome-wide siRNA screen uncovered a large number of potential regulators of signaling, but none of the knockdowns completely blocked adaptation, suggesting redundancy in adaptation mechanisms (*Deglincerti et al., 2015*). As an alternative approach, we used FRAP at different times after Activin treatment to better understand how cells modulate nuclear SMAD4 levels over the course of the signaling response (*Figure 2a–c*). In particular, we performed measurements before ligand treatment, after 1 hr of treatment when signaling peaks, and after 6 hr when signaling has adapted (*Figure 2c*). In all cases, we observed recovery on a timescale of minutes following nuclear photobleaching, which confirmed that SMAD4 continuously shuttles between the nucleus and cytoplasm even in the absence of ligand stimulation (*Figure 2d*). We also observed a reduced recovery rate after 1 hr of Activin treatment, consistent with previous work on TGFβ signaling which suggested that nuclear accumulation of SMAD4 is caused by reduced nuclear export (*Nicolás et al., 2004*; *Schmierer and Hill, 2005*). Importantly, however, we found that the recovery rate is not restored during adaptation (*Figure 2d*), showing that cells do not revert to a pre-signaling state. This excludes upstream mechanisms of adaptation as might be caused by secretion of extracellular feedback inhibitors or depletion of receptors or R-Smads (*Vizán et al., 2013*). Moreover, mathematical modeling showed that if adaptation is caused by depletion of an upstream component such as receptors, then the magnitude of adaptation is governed by the ratio of the degradation rates of active and inactive receptors (Appendix 1). These degradation rates also control the timescales for adaptation and recovery so that strong adaptation necessitates a large difference in timescales. However, we do not see these different time scales in our pulse experiments below (see Figure 5), which show that the refractory period after ligand exposure is similar to the time scale of adaptation. In contrast, models with negative feedback causing inhibition of downstream signaling are capable of explaining all features of our data including the observed time scales (Appendix 1).

To understand the mechanism of this inhibition, we first considered fitting our FRAP data to a model in which the entire population of SMAD4 is free to shuttle between the nucleus and cytoplasm. In this case, bleaching reduces the total pool of observable GFP-SMAD4 molecules but the nuclear to cytoplasmic intensity ratio depends only on the kinetic constants, and therefore would recover to the same value following bleaching. However, we observed that this ratio systematically decreases after nuclear bleaching (*Figure 2f*). This suggests a more general model that includes sequestered SMAD4, which may move within but not between the nuclear and cytoplasmic compartments. Because production and degradation are slow compared to shuttling, recovery from bleaching comes only from redistribution of unbleached molecules. The nuclear sequestered population is not exported to the cytoplasm, while the cytoplasmic sequestered population is unavailable to enter the nucleus and replenish the population of unbleached molecules there, leading to a reduced nuclear to cytoplasmic ratio after equilibration of the free dark and fluorescent molecules through exchange (*Figure 2g,h*, Appendix 1). The parameters obtained from fitting this model to our data suggest that initial accumulation of SMAD4 in the nucleus reflects both a lower export rate and a reduction in cytoplasmic sequestration (*Figure 2i,j*). Adaptation, however, does not result from altering shuttling kinetics, but instead reflects reduced nuclear sequestration and increased cytoplasmic sequestration. Thus, taken together, our FRAP data and mathematical modeling suggest a

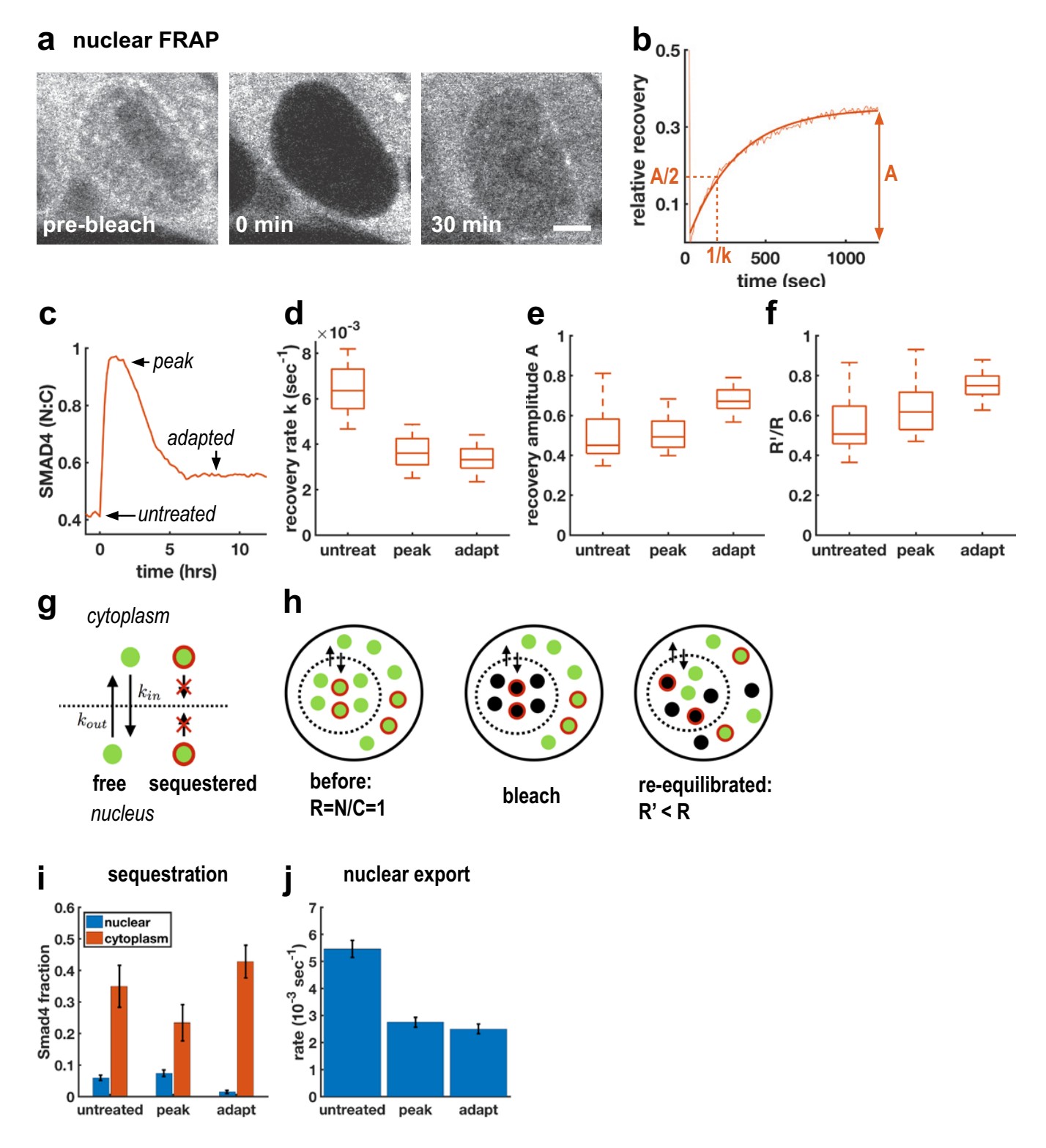

**Figure 2.** Adaptive Activin response is not a return to the pre-stimulus state and is explained by a model that includes sequestered SMAD4 populations. (a) Photobleaching and recovery of nuclear Smad4 at 2 hr after Activin treatment. Scalebar 5 μm. (b) Exponential fit to recovery of nuclear fluorescence after bleaching yields amplitude A and recovery rate k. Intensity drop at t = 0 shows bleaching event. (c) Photobleaching was performed on untreated cells, at the peak response to Activin, and after adaptation. (d) Boxplot of distribution of recovery rates. Recovery rates at peak signaling and after adaptation are significantly smaller than for untreated cells (t-test $p < 10^{-6}$), the difference in recovery rate between peak signaling and

*Figure 2 continued on next page*

*Figure 2 continued*

adapted state is not significant. N > 12 cells for each FRAP condition. (**e**) Boxplot of distribution of recovery amplitudes. (**f**) Nuclear to cytoplasmic intensity ratio after bleaching (R') is systematically smaller than nuclear to cytoplasmic intensity ratio before bleaching (R). (**g**) Cartoon of mathematical model for Smad4 localization, with sequestered populations of Smad4 that are confined to either nucleus or cytoplasm, and free Smad4 shuttling with import/export rates $k_{in}/k_{out}$. (**h**) Cartoon demonstrating that this model explains results in (**f**). (**i**) Changes in Smad4 sequestration in nucleus (blue) and cytoplasm (red) determined through model fitting. Error bars in i and j represent error propagation of standard errors in measured parameters over N > 12 cells as described in Appendix 1. (**j**) Nuclear export rates, which given the fixed nuclear import rate of the model directly reflect the measured exchange rates k.

DOI: https://doi.org/10.7554/eLife.40526.007
The following source data is available for figure 2:
**Source data 1.** MATLAB script and .mat files to reproduce the data panels in *Figure 2*.
DOI: https://doi.org/10.7554/eLife.40526.008

mechanism of adaptation that relies on negative feedback and acts by modulating sequestration rather than nuclear exchange rates.

## Transcriptional dynamics of differentiation targets follows SMAD4 dynamics

Next, we evaluated transcriptional dynamics downstream of BMP4 and Activin using qPCR, which showed that BMP targets are stably induced (*Figure 3a*), while differentiation targets of Nodal show adaptive transcription on a timescale consistent with SMAD4 signaling (*Figure 3b*). Moreover, shared targets of the pathways were found to be transcribed adaptively in response to Activin and stably in response to BMP4 (*Figure 3c*, *Figure 3—figure supplement 1a*). In contrast, the transcription of *NODAL*, *WNT3*, and their inhibitors *LEFTY1* and *CER1* were sustained upon Activin treatment (*Figure 3d*). Molecularly, the two classes of transcriptional dynamics in response to Activin may reflect differential requirements for SMAD4 signaling levels with lower levels required to maintain the targets with sustained dynamics so that these are continuously transcribed due to the baseline signaling following adaptation. Alternatively, transcription of these genes may require only SMAD2/3 activation, which is more sustained than that of SMAD4 (*Figure 1—figure supplement 1*e,g,h). The differences in expression of these sets of targets are not due to differences in mRNA stability as mRNAs for stably expressed genes were found to decline rapidly upon pathway inhibition with SB431542 indicating a need for ongoing signaling to maintain expression (*Figure 3—figure supplement 1g*).

The sustained transcription of Nodal and Wnt pathway ligands and inhibitors may be required to activate the positive feedbacks between the Nodal and Wnt pathways, which are known to be involved in establishing the primitive streak, the region of the mammalian embryo where mesoderm and endoderm form (*Ben-Haim et al., 2006*). This suggests a picture where stable transcription of the ligands and inhibitors allows for the establishment of signaling patterns in the embryo, while cells receiving these signals to differentiate interpret them according to their dynamics. Several other genes not related to mesendoderm differentiation were also found to be stably induced by Activin (*Figure 3—figure supplement 1b–d*).

The measurements above were performed by adding Activin to mTeSR1 media which contains high levels of FGF. Activin/Nodal signaling plays a dual role in the early embryo and is involved in both maintaining pluripotency, and differentiation to primitive streak fates. It maintains epiblast pluripotency in combination with FGF (*James et al., 2005*; *Vallier et al., 2005*), and while differentiation genes are transiently induced by Activin treatment in the presence of only FGF, cells do not robustly differentiate. In contrast, Activin induces primitive streak fates when Wnt is present. We asked whether target genes also respond adaptively to Activin/Nodal signaling during this differentiation. Initial transcriptional response was found to be qualitatively similar with or without Wnt activation, although shared targets of Wnt and Activin such as the mesendodermal markers *EOMESODERMIN* (*EOMES*) and *BRACHYURY* (*BRA*) showed a stronger response and were stably reactivated following adaptation on longer time scales (*Figure 3e,f*, *Figure 3—figure supplement 1e,f*).

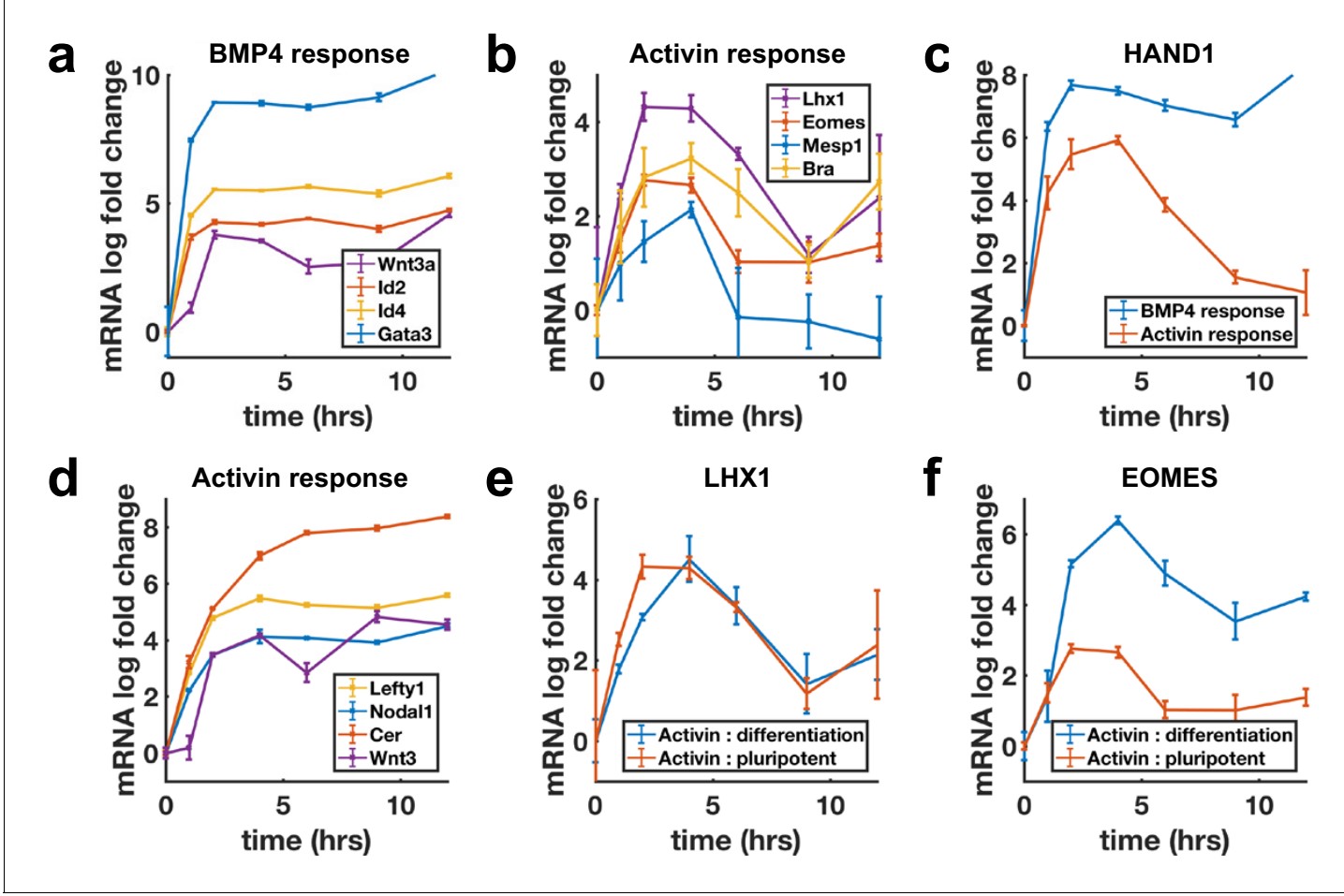

**Figure 3.** Transcription of BMP targets and Nodal differentiation targets reflects SMAD4 dynamics, while other Nodal targets show sustained transcription. (a, b) qPCR measurements of transcriptional response to BMP4 treatment (a) and of differentiation targets to Activin (b) y-axes show relative $C_T$ values. (c) Transcription of the shared Activin/BMP4 target *HAND1* after BMP4 (blue) or Activin (red) treatment. (d) Non-adaptive response to Activin of ligands and inhibitors involved in initiating the primitive streak. (e) Transcriptional response to Activin under pluripotency maintaining conditions (red) and mesendoderm differentiation conditions (blue) of Activin target *LHX1* (e) and joint Activin/Wnt target *EOMES* (f). Error bars represent standard deviations over three replicates. Logarithms are base 2.

DOI: https://doi.org/10.7554/eLife.40526.009

The following source data and figure supplement are available for figure 3:

**Source data 1.** MATLAB script and .mat files to reproduce the data panels in *Figure 3*.
DOI: https://doi.org/10.7554/eLife.40526.011
**Figure supplement 1.** Additional qPCR data.
DOI: https://doi.org/10.7554/eLife.40526.010

## BMP4 response reflects concentration, Activin response reflects rate of concentration increase

Sustained response to BMP4 suggests sensing of ligand concentration. In contrast, adaptive response to Activin with dose-dependent amplitude suggests sensing of ligand rate of increase. We directly tested whether cells are sensitive to the rate of increase of Activin, but not BMP4, by slowly raising ligand concentrations (concentration ramp) and comparing the response with the response to a single step to the same final dose (*Figure 4a,d*). If cells are primarily sensitive to ligand doses, the step and ramp should eventually approach the same final activity, while if cells are sensitive to the rate of ligand increase, the response to the ramp should be reduced. As expected, BMP4 signaling responses to the ramp and step approached each other as ramp concentration increased, while Activin signaling was dramatically reduced in the case of the ramp (*Figure 4b,e*). Moreover,

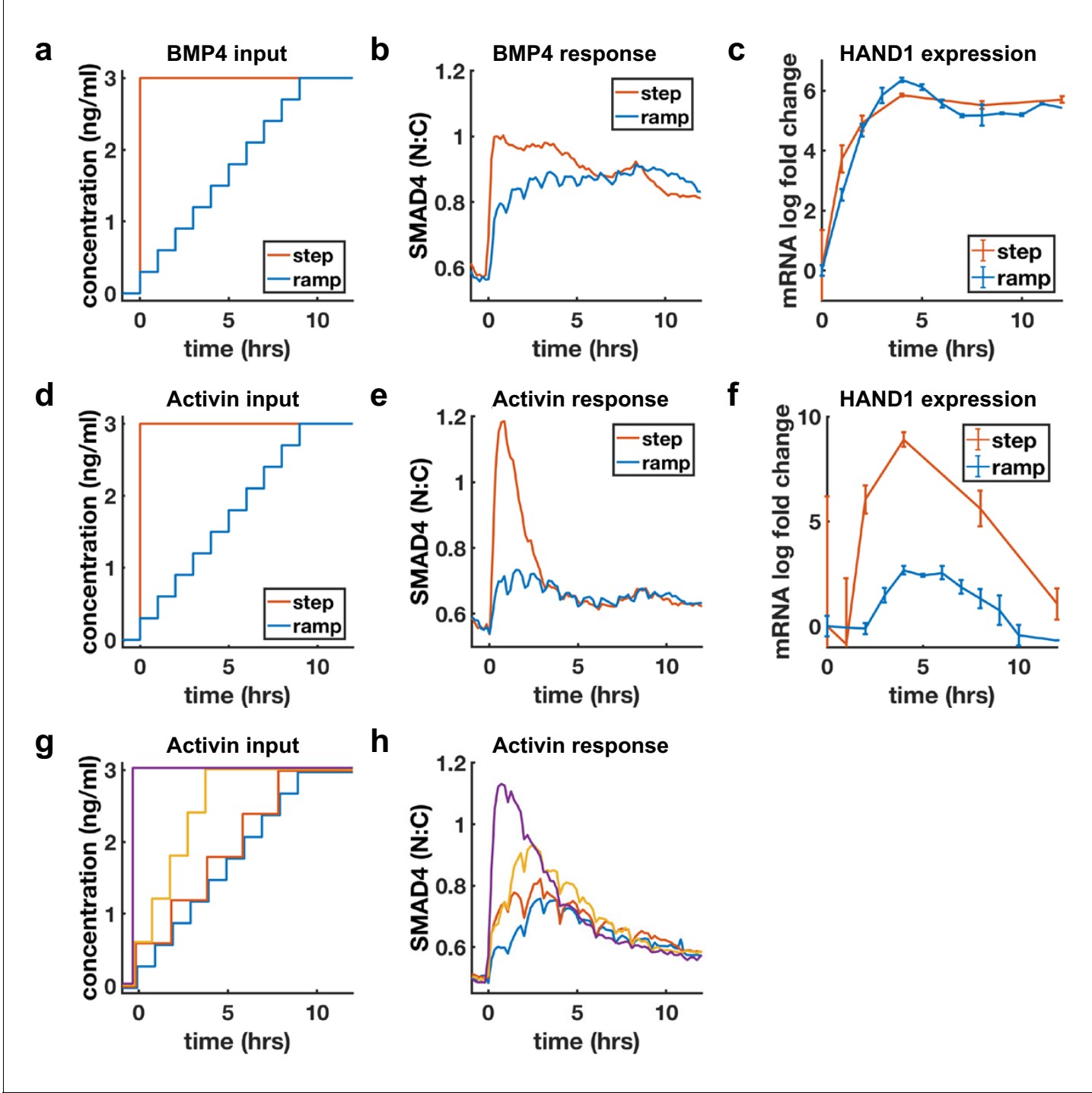

**Figure 4.** BMP4 response reflects concentration, but Activin response reflects rate of concentration increase. (**a, d**) Ligand concentration over time for slow ramp (blue) or sudden step (red) in BMP4 (**a**) or Activin (**d**). (**b, e**) SMAD4 signaling response to BMP4 (**b**) or Activin (**e**). (**c, f**) Transcriptional response of *HAND1* to concentration ramp versus step for BMP4 (**c**) or Activin (**f**). Error bars in qRT-PCR data (**c, f**) represent standard deviations over three replicates. (**g, h**) Activin response to different ramp rates and step sizes. In (**b, e, h**) small hourly wiggles are artifacts of performing media changes and do not reflect actual signaling responses.

DOI: https://doi.org/10.7554/eLife.40526.012

The following source data and figure supplement are available for figure 4:

**Source data 1.** MATLAB script and .mat files to reproduce the data panels in *Figure 4*.

DOI: https://doi.org/10.7554/eLife.40526.014

*Figure 4 continued*

**Figure supplement 1.** Additional qPCR ramp data.
DOI: https://doi.org/10.7554/eLife.40526.013

transcriptional dynamics of the shared target *HAND1* matched the signaling pattern and showed dramatically reduced transcription in response to the Activin but not the BMP ramp (*Figure 4c,f*). As noted in the discussion of *Figure 1e*, the BMP response is switch-like with increasing dosage, and this was reflected by the ramp approaching the levels of the step within the first few increases. Similar results were obtained for other adaptive Activin targets, in contrast to non-adaptive Activin targets which, as expected, also showed sustained transcription in response to the ramp (*Figure 4— figure supplement 1*). Finally, we varied the rate of the ramp of Activin and found that intermediate rates of increase also yielded intermediate signaling responses (*Figure 4g,h*).

## Repeated rapid increases in Activin/Nodal enhance differentiation to primitive streak fate

Morphogens control cell fate, and the dependence of transcription of mesodermal and endodermal genes on Activin rate of increase suggests rapid Activin increase may boost differentiation to these fates. We hypothesized that exposing cells to repeated rapid increases by pulsing the level of Activin could enhance this effect. To rigorously test this hypothesis, we grew cells in differentiation conditions and compared pulses that switch between high and low doses of Activin with a sustained high Activin dose while performing media changes at the same times ('dummy pulses') (*Figure 5a*). Each pulse of Activin elicited a strong response in the translocation of SMAD4 to the cell nucleus, while no such responses were seen in response to the dummy pulses (*Figure 5b*). Differentiation to mesendooderm as marked by Bra expression was also enhanced under pulsed conditions (*Figure 5c,d*), despite the reduced integrated ligand exposure. Continuous exposure to lower doses of Activin and treating cells with Activin for the same total time as the three pulses but in a single pulse showed that the effect is specific to pulsed Activin and not a consequence of simply reducing the integrated Activin exposure (*Figure 5e–g*, *Figure 5—figure supplement 1*).

## Rapid changes in endogenous Nodal signaling occur during self-organized patterning

To test whether rapid concentration increases are relevant to endogenous Nodal during embryonic patterning, we turned to micropatterned colonies of hESCs treated with BMP4. These colonies differentiate in a spatial pattern with reproducible rings of extraembryonic cells and all three germ layers, and represent a model for the patterning associated with gastrulation in the human embryo (*Heemskerk and Warmflash, 2016*; *Warmflash et al., 2014*). Nodal signaling is required for the formation of mesoderm and endoderm within these colonies, and both small molecule inhibition and genetic knockout of Nodal drastically reduce mesendoderm differentiation (*Chhabra et al., 2018*; *Warmflash et al., 2014*). For the dynamics described here to be relevant, Nodal signaling should evolve rapidly compared to the timescale for adaptation, and we should observe rapidly changing signaling patterns with the GFP:SMAD4 reporter. In contrast, if cells read a stable gradient of Nodal protein during patterning, as posited by classic models, we would expect correspondingly stable patterns in signaling and the adaptive dynamics would not be relevant.

To distinguish these hypotheses, we used live imaging to observe SMAD4 signaling in micropatterned colonies during the 42 hr in which these patterns form (*Figure 6*a-c). The cells initially respond uniformly to the BMP4 treatment. The response then is restricted to the colony edge by approximately 12 hr (*Figure 6a,b*), and this pattern is maintained until approximately 25 hr. Beginning at 25 hr, a wave of increased nuclear SMAD4 spreads inward again from the edge (*Figure 6a,c*). SMAD4 convolves the BMP and Nodal responses, and we hypothesized that the stable response at the edge of the colony represents BMP signaling while the inward moving wave results from Nodal signaling. To test this, we looked at the pathway specific SMAD1 and SMAD2/3 (*Figure 6*d-g). These confirmed that the BMP-specific active SMAD1 signaling remains restricted to the edge (*Figure 6f*), while the Nodal transducer SMAD2/3 activity spreads rapidly inward from the colony edge between 24 and 36 hr (*Figure 6e*). The SMAD2/3 and SMAD4 signal transducers reveal a rapidly evolving Nodal

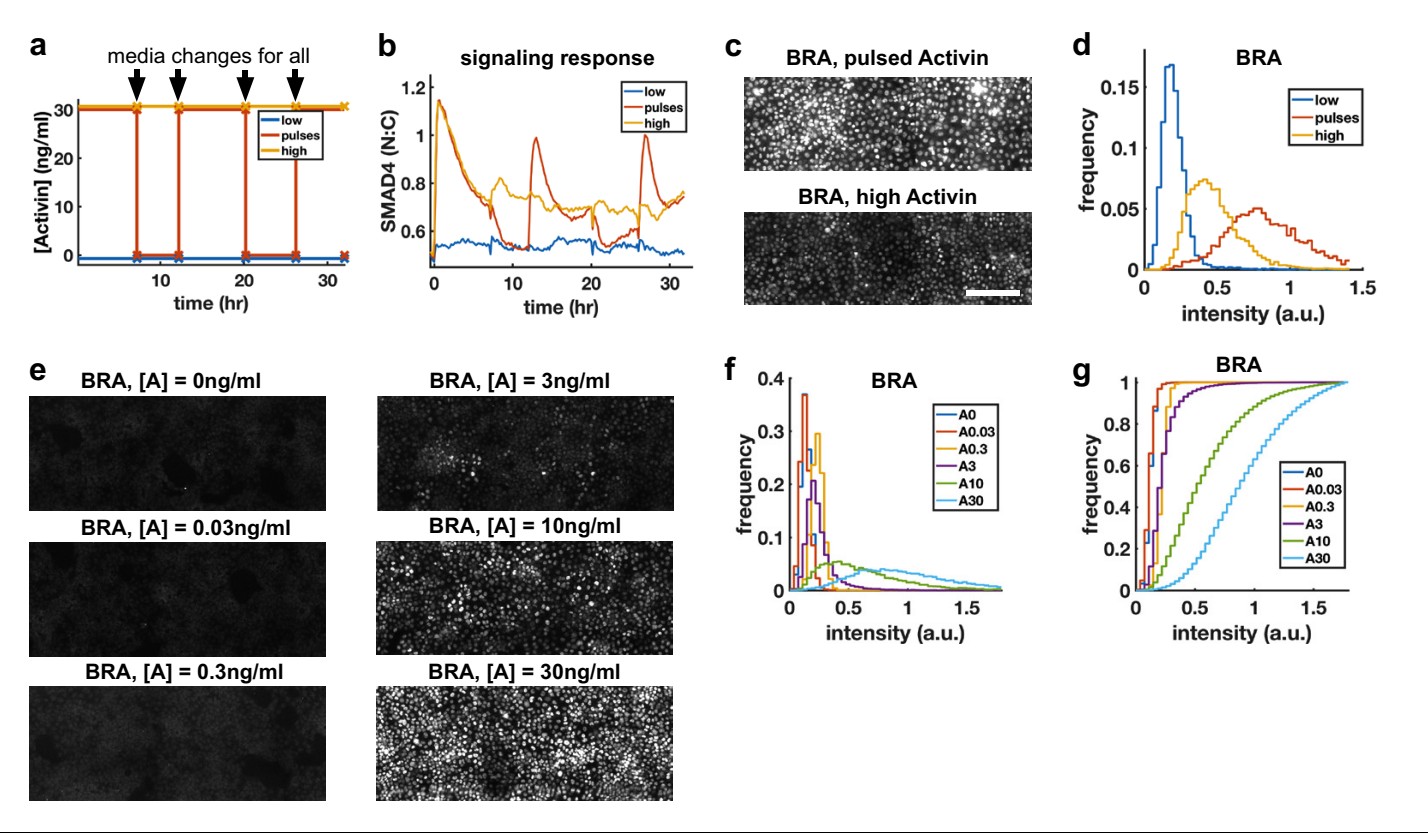

**Figure 5.** Repeated rapid increases in Activin/Nodal enhance differentiation to primitive streak fate. (a) Schematic of pulse experiment, graph shows ligand concentration, controls receive media changes at the same time as the pulsed well. (b) SMAD4 signaling profile in response to Activin pulses (red), high Activin (yellow), and no Activin (blue). (c) Immunofluorescence staining for BRA after high constant Activin or pulsed Activin. Scalebar 100 µm. (d) Distribution of BRA expression per cell ($N_{cells}$ per condition ~6 × 10³) determined from immunofluorescent images (c). (e, f, g) Dose response series showing BRA expression monotonically increases with Activin dose, and therefore the effect of pulses is not due to reduced average Activin exposure. (e) Immunofluorescence staining for BRA after 34 hr differentiation with different doses of Activin. (f) Distributions of BRA intensity per cell in the images containing (d). (g) Cumulative distributions of BRA intensity.

DOI: https://doi.org/10.7554/eLife.40526.015

The following source data and figure supplement are available for figure 5:

**Source data 1.** MATLAB script and .mat files to reproduce the data panels in *Figure 5*.
DOI: https://doi.org/10.7554/eLife.40526.017
**Figure supplement 1.** Increased BRA expression after pulsing is not due to reduced integrated exposure.
DOI: https://doi.org/10.7554/eLife.40526.016

distribution with a wavefront that moves through the colony at approximately one-cell diameter (10 µm) per hour. The velocity of the Nodal wavefront suggests that individual cells see Nodal levels increase rapidly in 1 hr, consistent with the time scale of ligand increase required for strong response to exogenous ligand in the previous experiments. Importantly, this wave of signaling activates Bra expression in its wake (*Figure 6g,h*). This indicates that rapid increases in endogenous Nodal signaling are associated with mesoderm differentiation in this model of human gastrulation.

## Discussion

Our work shows that morphogens in the mammalian embryo do not act in a purely concentration-dependent matter and has revealed an important role for Activin/Nodal rate of change in specifying cell fate as a consequence of adaptive signaling. Adaptive signaling could serve to restrict the response to a narrow competence window or to separate the multiple roles of a single morphogen by using distinct dynamics to selectively activate different target genes, effectively expanding the

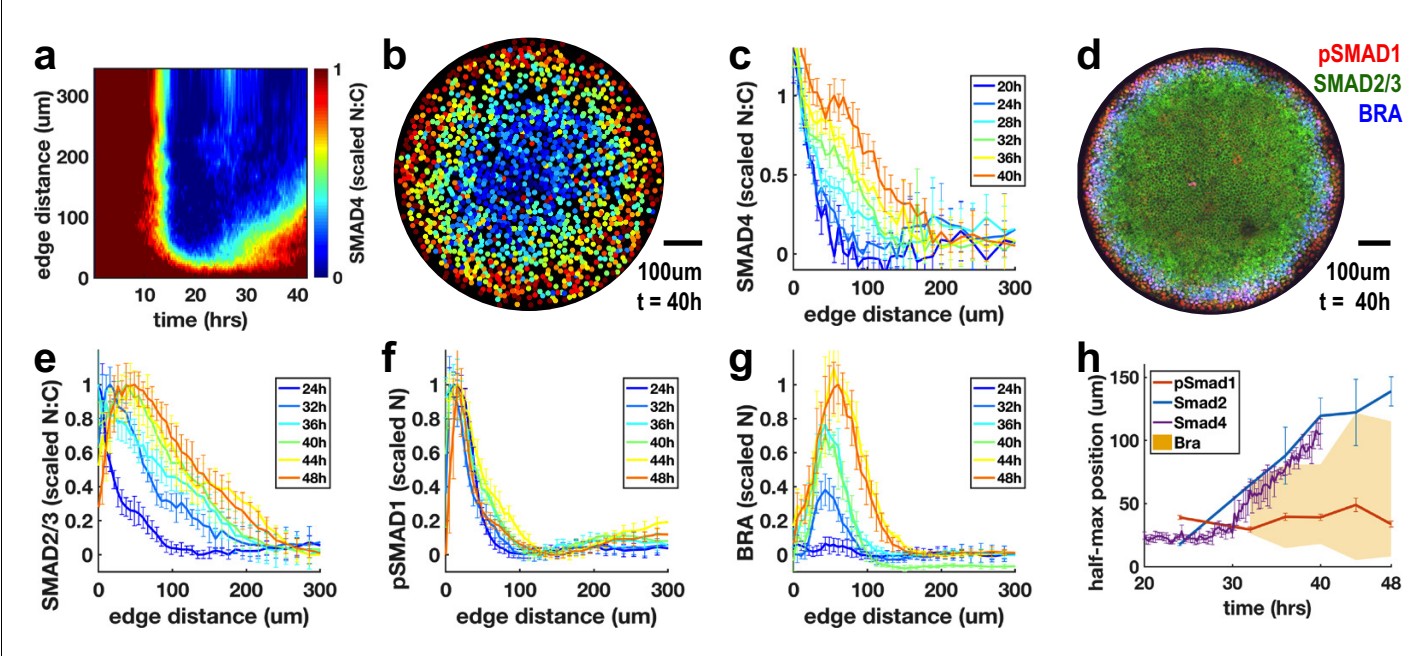

**Figure 6.** Rapid changes in endogenous Nodal signaling occur during self-organized patterning. (**a**) Average radial profile of SMAD4 signaling over time (kymograph) in micropatterned colonies after BMP4 treatment (N = 4 colonies). (**b**) SMAD4 signaling in single colony at 40 hr. (**c**) Radial SMAD4 signaling profiles at discrete times from 20 hr to 40 hr. (**d**) Immunofluorescence staining 40 hr after BMP4 treatment for pSMAD1, SMAD2/3 and BRA. (**e, f, g**) Normalized radial profiles of SMAD2/3 (**e**) pSMAD1 (**f**) and BRA (**g**) averaged over N > 5 colonies per time. (**h**) Half-maximum versus time for SMAD2/3 (blue), SMAD4 (purple) and pSMAD1 (red) and BRA expression domain defined by a threshold of at least 20% of maximal expression (yellow). In all panels, error bars represent standard deviations taken over different colonies.

DOI: https://doi.org/10.7554/eLife.40526.018

The following video and source data are available for figure 6:

**Source data 1.** MATLAB script and .mat files to reproduce the data panels in *Figure 6a–c*.
DOI: https://doi.org/10.7554/eLife.40526.020
**Source data 2.** Additional data for *Figure 6—source data 1*.
DOI: https://doi.org/10.7554/eLife.40526.021
**Source data 3.** MATLAB script and .mat files to reproduce the data panels in *Figure 6e–h*.
DOI: https://doi.org/10.7554/eLife.40526.022
**Figure 6—video 1.** SMAD4 signaling in a micropatterned colony reveals Nodal signaling wave.
DOI: https://doi.org/10.7554/eLife.40526.019

information content of the morphogen gradient (*Selimkhanov et al., 2014*). It is currently unclear whether adaptive signaling is an intrinsic feature of the Activin/Nodal pathway or is context dependent. A recent study found that Activin target genes are induced more stably when cells are also treated with Wnt, but that the dynamics of Smad2 translocation were not affected (*Yoney et al., 2018*). For Wnt, costimulation with Activin or BMP switches the dynamics of Wnt signaling from transient to sustained, and it will be interesting to determine whether Activin/Nodal signaling is sustained in some contexts (*Massey et al., 2019*).

Activin and Nodal are generally thought of as text-book morphogens acting in a concentration-dependent manner (*Gilbert, 2014*). This is based on experiments in which an artificial gradient is created by placing an Activin-soaked bead in an intact animal cap or in which dissociated Xenopus animal cap cells are exposed to a range of Activin concentrations (*Green et al., 1992*; *Gurdon et al., 1994*). It is important to note that these experiments are not inconsistent with our findings. In both cases, the highest concentration corresponds to the highest rate of concentration increase, and our results also show the peak of the transient signaling response induced by a step increase is dose-dependent. This fact could underlie the differences in cell fates observed in dissociated animal cap cells.

Our results are in contrast to those for another well-characterized system, the patterning of the neural tube by Shh, which has also been shown to be adaptive and regulates cell fate decisions through a described gene regulatory network (*Balaskas et al., 2012*). In that case, adaptation depends on upstream inhibition and results in a relatively constant integrated signaling output in which time and duration can be interchanged (*Dessaud et al., 2007*). This interchangeability between concentration and duration can be achieved by upstream feedback such as degradation of activated receptors. Intuitively, increasing ligand concentration increases the response but also leads to more rapid degradation of receptors limiting the response in time.

Several lines of reasoning argue that adaptation in Activin/Nodal signaling follows a different mechanism, with consequences for cell fate. The signaling behavior of our system is qualitatively different, as there is no tradeoff between concentration and duration of strong signaling. Longer exposure to ligand does not increase the duration of signaling because the system has already adapted, and reactivating the pathway can only be achieved by pulsing rather than extending the duration of ligand exposure. Interestingly, there is a low level of baseline signaling following adaptation which is maintained by continued ligand exposure. Given the distinct classes of target genes we found responding to each aspect of the signal (baseline and adaptive pulse), this may demonstrate the principle that dynamic signals relay more information through a single pathway, with one set of target genes responding to signal duration, and another to rapid signal increase.

In addition to the qualitative behavior being inconsistent with a system that integrates signal, like the neural tube, our quantitative data and mathematical modeling rule out the mechanism that would most naturally implement such a behavior, namely adaptation through degradation. First, our FRAP measurements show that the adapted state is kinetically distinct from the pre-signaling state, while the degradation model would predict a return to the same state. Second, we show analytically in Appendix 1 that a model based on upstream feedback cannot account for our dynamic measurements because in order to adapt through receptor degradation, the timescale for recovery must be slower than that for adaptation, which is not what we find in our pulsing experiments. It will be interesting to elucidate the genetic network that interprets adaptive Nodal signaling and to compare it with the one that interprets Shh.

Our conclusions regarding signaling dynamics rely on nuclear localization of Smads as a proxy for signaling activity. The validity of this measure is supported by strong correlation between SMAD2 nuclear localization and pSMAD2 levels, as well as the fact that transcriptional activity of a number of direct targets mirrors nuclear SMAD4 both in this paper and in previous work (*Warmflash et al., 2012*). Although some target genes do not follow this time course, it is important to note that target gene dynamics represent those of signaling filtered through a specific promoter (*Dubrulle et al., 2015*). For example, a promoter with high affinity for the signal transducer will saturate at low levels and will not reflect fluctuations in signaling that remain above these levels. Although it was shown that blocking nuclear export of Smad4 does not interfere with function of the pathway (*Biondi et al., 2007*), this does not invalidate our conclusions, as blocking nuclear export of Smad4 destroys the strong correlation between pathway activity and nuclear localization by artificially keeping Smad4 in the nucleus following the termination of transcription.

Although relatively unexplored in the context of development, adaptive signaling is a common feature of biological systems. Well-studied examples in bacteria include the chemotactic response where adaptive signaling allows cells to sense the rate of change and move to areas of higher nutrients (*Block et al., 1983*), and the $\sigma_b$ response where it allows cells to sense the rate of increase of environmental stress (*Young et al., 2013*). In mammalian cells, stimulation with TNF$\alpha$ leads to a transient pulse of NF$\kappa$B signaling, followed by a low, sustained baseline, and this has been suggested to allow specificity in target gene activation with different targets responding to different features of the dynamics (*Hoffmann et al., 2002*). Similarly, DNA damage has been shown to give rise to stereotyped pulses of p53 signaling due to negative feedback (*Lahav et al., 2004*).

It will be important to understand how our results fit with current protocols for directed stem cell differentiation which typically perform a single media change per day and therefore do not take advantage of potential enhancements resulting from optimizing ligand dynamics. Routinely, ranges of ligand concentration are explored to optimize differentiation protocols. Our results suggest a different approach that starts with characterizing the signaling dynamics during each cell fate decision and then optimizing the relevant dynamics of ligand presentation. As such, we believe the results presented

here will serve as a basis for a dynamic understanding of embryonic patterning and stem cell differentiation.

# Materials and methods

### Key resources table

| Reagent type (species) or resource | Designation | Source or reference | Identifiers | Additional information |
|---|---|---|---|---|
| Cell line (*Homo sapiens*) | ESI017 | ESIBIO | RRID:CVCL_B85 | |
| Cell line (*H. sapiens*) | RUES2 | Ali Brivanlou (Rockefeller) | RRID:CVCL_B810 | |
| Antibody | Goat polyclonal anti-Brachyury | R and D Systems | RRID:AB_2200235 | (1:400) |
| Antibody | Rabbit monoclonal anti-Sox2 | Cell Signaling Technologies | RRID:AB_1904142 | (1:200) |
| Antibody | Mouse monoclonal anti-Nanog | BD Biosciences | RRID:AB_1645598 | (1:400) |
| Antibody | Mouse monoclonal anti-Smad2/3 | BD Biosciences | RRID:AB_398162 | (1:100) |
| Antibody | Rabbit monoclonal anti-pSmad1 | Cell Signaling Technologies | RRID:AB_2493181 | (1:100) |
| Software, algorithm | Image processing and data analysis code | This study | https://github.com/idse/stemcells/ commit a5ee164 | |

### Cell lines

The cell lines used were ESI017 and RUES2 GFP:Smad4 RFP:H2B. ESI017 cells were obtained directly from ESIBIO while RUES2 were a gift of Ali Brivanlou (Rockefeller University). The identity of these cells as pluripotent cells was confirmed via triple staining for pluripotency markers OCT4, SOX2, and NANOG. All cells were routinely tested for mycoplasma contamination and found negative.

### Cell culture and differentiation protocols

The cell lines and their maintenance are described in *Nemashkalo et al. (2017)*. For all experiments except micropatterning, cells were seeded at a low density of $6 \times 10^4$ cells per cm$^2$ and grown with rock-inhibitor Y27672 (10 uM; StemCell Technologies). This ensured uniform response to exogenous ligand and minimized the effect of secondary endogenous signaling. Unless otherwise indicated in the figures, experiments in *Figures 1–3* were done in mTeSR1 medium (StemCell Technologies) (also referred to as pluripotency conditions), and cells were treated with 50 ng/ml Activin A (R and D Systems) or 50 ng/ml BMP4 (R and D Systems). Noggin was used at 500 ng/ml. SB431542 at 10 µM. Differentiation conditions in *Figure 3*e,f are defined as Essential six medium (Gibco) +3 µM CHIR99021 (StemCell technologies).

### Imaging and image analysis

Live imaging was done on an Olympus/Andor spinning disk confocal microscope with a 40×, NA 1.25 silicon oil objective. Immunofluorescence data for *Figure 5* were collected using a 20×, NA 0.75 objective on an Olympus IX83 inverted epifluorescence microscope. Fixed micropatterned colonies for *Figure 6* were imaged using an 30x NA 1.05 silicon oil objective on an Olympus FV1200 laser scanning confocal microscope. All image analysis used Ilastik (*Sommer et al., 2011*) for initial segmentation and custom written MATLAB code, available at https://github.com/idse/stemcells for further analysis (*Heemskerk, 2019*; copy archived at https://github.com/elifesciences-publications/stemcells). *Figure 1—figure supplement 1a* shows how Smad4 nuclear to cytoplasmic ratio, our measure for signaling intensity, is defined from the segmentation by subtracting the mean value in the background mask from nuclear and cytoplasmic masks for each cell. Smad2/3 signaling was

similarly measured by nuclear to cytoplasmic ratio. Since pSmad1 and Bra are purely nuclear, these were measured by nuclear intensity, normalized by DAPI to correct for intensity variation due to optics.

## FRAP

FRAP experiments were done on an Olympus FV1200 laser scanning confocal microscope using a 100x NA 1.49 oil objective. To deal with cell movement recovery, curves were obtained by reading out a polygon defined by the interpolation between the initial bleach window and a manually defined polygonal nuclear mask in the final frame. Supplemental Information describes the mathematical model for nuclear localization of Smad4 which is shown schematically in *Figure 2g,h*, and the inference of model parameters shown in *Figure 2i,j* from the data.

## qPCR

For qPCR experiments, ESI017 cells were grown in 24-well plates. RNA was extracted using Ambion RNAqueous-Micro Total RNA Isolation Kit and cDNA synthesis was performed with Invitrogen Super-Script Vilo cDNA Synthesis Kit according to the manufacturer's instructions. Measurements were performed with SYBR green and the primers in the *Table 1*. ATP5O was used for normalization in all experiments.

## Pulses

For pulse and dose response experiments in *Figure 5*, differentiation was done in Essential six medium +1 uM CHIR99021 +20 ng/ml bFGF (Life Technologies)+10 uM Y27672 with 30 ng/ml Activin added as indicated in the figures. Time between pulses was always 5 hr to allow the pathway to relax. Duration of individual pulses was chosen for experimental convenience and lengths of pulses shown in *Figure 5* are 6, 10, and 8 hr. Controls were subjected to media changes at the same time. During media changes, cell were washed three times with PBS.

## Micropatterning

For micropatterned colonies, we followed the protocol in *Deglincerti et al. (2016)* using the chemically defined medium mTeSR1. For fixed micropatterns, we used the CYTOO Arena EMB chip and analyzed the 800 um colonies, while for live imaging we used a CYTOO 96-well plate RW DS-S-A,

**Table 1.** qPCR primers used in this study.

| ATP5O | ACTCGGGTTTGACCTACAGC | AAAATGAACGGACAGAACCG |
|---|---|---|
| BRA | TGCTTCCCTGAGACCCAGTT | GATCACTTCTTTCCTTTGCATCAAG |
| CER | ACAGTGCCCTTCAGCCAGACT | ACAACTACTTTTTCACAGCCTTCGT |
| GATA3 | TTCCTCCTCCAGAGTGTGGT | AAAATGAACGGACAGAACCG |
| HAND1 | GTGCGTCCTTTAATCCTCTTC | GTGAGAGCAAGCGGAAAAG |
| ID2 | GCAGCACCTCATCGACTACA | AATTCAGAAGCCTGCAAGGA |
| ID4 | CCCTCCCTCTCTAGTGCTCC | GTGAACAAGCAGGGCGAC |
| LEFTY1 | ACCTCAGGGACTATGGAGCTCAGG | AGAAATGGCCAATTGAAGGCCAGG |
| LHX1 | TCCCCAATGGTCCCTTCTC | CGTAGTACTCGCTCTGGTAATCTCC |
| MIXL1 | CCGAGTCCAGGATCCAGGTA | CTCTGACGCCGAGACTTGG |
| NANOG | CCGGTCAAGAAACAGAAGACCAGA | CCATTGCTATTCTTCGGCCAGTTG |
| NODAL | ATGCCAGATCCTCTTGTTGG | AGACATCATCCGCAGCCTAC |
| NOG | CATGAAGCCTGGGTCGTAGT | TCGAACACCCAGACCCTATC |
| OCT4 | GGGCTCTCCCATGCATTCAAAC | CACCTTCCCTCCAACCAGTTGC |
| SOX2 | CCATGCAGGTTGACACCGTTG | TCGGCAGACTGATTCAAATAATACAG |
| TBR2/EOMES | CACATTGTAGTGGGCAGTGG | CGCCACCAAACTGAGATGAT |
| WNT3 | CTCGCTGGCTACCCAATTT | GAGCCCAGAGATGTGTACTGC |

DOI: https://doi.org/10.7554/eLife.40526.023

**Table 2.** Antibodies used for immunofluorescence in this study.

| Protein | Species | Dilution | Catalog no. | Vendor |
|---------|---------|----------|-------------|--------|
| BRA | Goat | 1:400 | AF2085 | R and D Systems |
| SOX2 | Rabbit | 1:200 | 5024S | Cell Signaling Technology |
| NANOG | Mouse | 1:400 | 560482 | BD Biosciences |
| SMAD2/3 | Mouse | 1:100 | 610842 | BD Biosciences |
| pSMAD1 | Rabbit | 1:100 | 13820 | Cell Signaling Technology |

DOI: https://doi.org/10.7554/eLife.40526.024

which has 700 um colonies. For the analysis in *Figure 6*, radial profiles of SMAD1 and SMAD2/3 were normalized to have the lowest signaling level be zero and the highest be one in at each time, as minimal and maximal levels were similar at each time and only their spatial distribution varied. SMAD4 was normalized such that its maximum over all positions interior in the colony to the most interior half-maximum of pSMAD1 intensity was equal to one. This position was chosen for normalization as it reflects the peak of Nodal-dependent SMAD4 signaling. Finally, because BRA levels change substantially, BRA at all times was normalized to the maximum of the latest time (48 hr).

## Immunostaining
Fixing and immunostaining of cells was done as described in *Nemashkalo et al. (2017)*. *Table 2* lists the antibodies that were used.

## Replicates, sample sizes, and error bars
All experiments were performed at least twice. Data shown are from representative experiments, except for FRAP data, where data from multiple experiments are pooled because of the difficulty of performing FRAP on a sufficient number of cells in a given experiment. Error bars on single-cell data (SMAD4 signaling dynamics, pSMAD1, SMAD2/3, BRA, and HAND1 immunostainings) are standard error over cells. Full distributions are also provided to show cell-to-cell variability. Error bars on qRT-PCR data are over technical replicates due to the difficulty of quantitatively comparing biological replicates, likely owing to differences in culture densities, timing between seeding and ligand stimulation, and other culture variables, however, multiple biological replicates were performed in all cases. Error bars in micropatterning experiments are standard deviation over colonies. For single-cell imaging experiments, at least 600 cells were measured for each condition, for FRAP experiments at least 12 cells were measured for each condition, and for micropatterning experiments at least five colonies were analyzed for each condition.

## Supplemental information
Supplemental information includes four figures, three movies, and an Appendix on mathematical modeling.

## Acknowledgements
We thank Elena Camacho Aguilar for careful reading of the manuscript and Anastasiia Nemashkalo for providing data for *Figure 1h*. This work was supported by the National Science Foundation (grant MCB-1553228), the Cancer Prevention Research Institute of Texas (grant RR140073), the Longenbaugh Foundation, Rice University, and the Society in Science - Branco Weiss fellowship, administered by ETH Zurich.

## Additional information

### Funding

| Funder | Grant reference number | Author |
|--------|------------------------|--------|
| National Science Foundation | MCB-1553228 | Aryeh Warmflash |

| Cancer Prevention and Research Institute of Texas | RR140073 | Aryeh Warmflash |
| Gillson Longenbaugh Foundation | 15-0217 | Aryeh Warmflash |
| The Branco Weiss Fellowship – Society in Science | | Idse Heemskerk |

The funders had no role in study design, data collection and interpretation, or the decision to submit the work for publication.

### Author contributions
Idse Heemskerk, Conceptualization, Data curation, Software, Formal analysis, Supervision, Funding acquisition, Investigation, Visualization, Writing—original draft, Writing—review and editing; Kari Burt, Lizhong Liu, Investigation, Writing—review and editing; Matthew Miller, Investigation, Visualization; Sapna Chhabra, M Cecilia Guerra, Investigation, Methodology, Writing—review and editing; Aryeh Warmflash, Conceptualization, Supervision, Funding acquisition, Investigation, Writing—original draft, Project administration, Writing—review and editing

### Author ORCIDs
Idse Heemskerk (iD) http://orcid.org/0000-0002-8861-7712
Sapna Chhabra (iD) http://orcid.org/0000-0002-2479-0911
Aryeh Warmflash (iD) http://orcid.org/0000-0002-5679-2268

### Decision letter and Author response
Decision letter https://doi.org/10.7554/eLife.40526.031
Author response https://doi.org/10.7554/eLife.40526.032

# Additional files

### Supplementary files
• Transparent reporting form
DOI: https://doi.org/10.7554/eLife.40526.025

### Data availability
All data necessary for reproducing the figures as well as the scripts that produce the figures are provided for each figure as a. zip file. Image processing code is available from Github at https://github.com/idse/stemcells (copy archived at https://github.com/elifesciences-publications/stemcells).

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

## Appendix 1

DOI: https://doi.org/10.7554/eLife.40526.026

# 'Rapid changes in morphogen concentration control self-organized patterning in human embryonic stem cells'

## FRAP measurements and inference of model parameters

### Kinetic model

In this section we formulate a kinetic model for Smad4 localization and work out its consequences, the following sections will show that the assumptions on which our model is based are consistent with the data.

The simplest model for Smad4 involves a single population of freely shuttling Smad4 whose total $T_f$ changes slowly compared with the time scale of shuttling and whose localization is determined entirely by nuclear import and export (*Schmierer and Hill, 2005*). The nuclear free Smad4 $N_f$ and cytoplasmic free Smad4 $C_f$ then obey

$$\dot{N}_f = k_{in}C_f - k_{out}N_f, \quad T_f = N_f + C_f \quad \dot{C}_f = -\dot{N}_f, \tag{1}$$

where $k_{in}$ and $k_{out}$ denote the nuclear import and export rate, respectively. Combining these equations we obtain

$$\dot{N}_f = k_{in}T_f - kN_f, \quad k \equiv k_{in} + k_{out}, \tag{2}$$

from which we see that the system approaches equilibrium exponentially with a time scale $1/k$. In equilibrium, the free populations are given by

$$N_f^{eq} = \frac{T_f k_{in}}{k_{in} + k_{out}} = \kappa T_f, \quad C_f^{eq} = \frac{T_f k_{out}}{k_{in} + k_{out}} = (1 - \kappa)T_f, \quad \kappa \equiv \frac{k_{in}}{k_{in} + k_{out}} \tag{3}$$

We therefore see that in this model, the nuclear to cytoplasmic ratio is given by

$$\frac{N}{C} = \frac{\kappa}{1 - \kappa} = \frac{k_{in}}{k_{out}}, \tag{4}$$

which is independent of the amount of Smad4 and therefore unaffected by bleaching. In other words, the prediction is that after recovery from photobleaching the nuclear to cytoplasmic intensity will have the same value as before bleaching. This is not what we find (*Figure 2f*).

We therefore consider a more general model in which we assume that nuclear and cytoplasmic Smad4 consist of both free and sequestered populations

$$N = N_f + N_s, \quad C = C_f + C_s, \tag{5}$$

and where the free part cycles between nucleus and cytoplasm. We define sequestered as unable to shuttle between nucleus and cytoplasm, but possibly still free to diffuse within the compartment. It is important that this is different from the definition in *Schmierer and Hill (2005)* where sequestration is taken to imply reduced mobility within each compartment. We are agnostic about the molecular mechanism of sequestration. Moreover, slow shuttling of the sequestered populations and slow interconversion between free and sequestered (i.e. changes in $T_f$) do not affect the conclusions as long as the time scales are long compared to the time scale of shuttling.

What we measure is fluorescence intensity which corresponds to Smad4 density, related to total Smad4 by the volume $N = nV_n$, $C = cV_c$, so the measured nuclear to cytoplasmic ratio is given by

$$r \equiv \frac{n}{c} = \alpha R = \alpha \ \frac{N_s + \kappa T_f}{C_s + (1-\kappa)T_f}, \qquad \alpha \equiv \frac{V_c}{V_n} \tag{6}$$

Measurements show that total Smad4 turns over very slowly (half life greater than 12 hr), so we can still consider it fixed on the time scale of our experiments

$$T = T_f + N_s + C_s. \tag{7}$$

Moreover, we are only interested in the relative size of the Smad4 populations, so we can set $T = 1$, and end up with the equilibrium nuclear to cytoplasmic ratio

$$r = \alpha \ \frac{N_s + \kappa(1 - N_s - C_s)}{C_s + (1-\kappa)(1 - N_s - C_s)}. \tag{8}$$

If nuclear exchange is much faster than the rate of change of any of the parameters in the expression above, we will always find the system in equilibrium and we can understand changes in nuclear to cytoplasmic ratio through the above expression. Since we do not expect $\alpha$ to change rapidly in response to Activin or BMP4, we conclude that an increase in $r$ can be achieved through increase in $N_s$, decrease in $C_s$, increase in $k_{in}$, or decrease in $k_{out}$. As we will show in the next section, this model has a enough freedom to fit our data and makes some basic predictions that we verified.

## Fluorescence recovery after photobleaching

To determine the kinetics of Smad4 shuttling, we used FRAP, which turns part of the Smad4 dark, allowing us to observe the equilibration dynamics of the complement. Nuclear bleaching initially sets the observable part of $N = 0$, while cytoplasmic bleach sets $C = 0$. Denoting quantities after bleaching with a prime, in the case of nuclear bleach we have $N_s' = 0$, while the new total free part $T_f'$ is the original cytoplasmic free part $T_f' = C_f = (1-\kappa)T_f$. Substituting this into the expression in the previous section yields the new equilibrium quantities and from (6) yields the prediction that the nuclear to cytoplasmic ratio goes down after nuclear bleach if there is sequestered Smad4

$$R' = \frac{N'}{C'} = \frac{A_N}{1 - A_C} R = \frac{\kappa(1-\kappa)T_f}{C_s + (1-\kappa)^2 T_f} \leq R \tag{9}$$

This inequality becomes an equality only when both $C_s = 0$ and $N_s = 0$. The approach to equilibrium is found by solving (2). Normalizing by the pre-bleach level, the nuclear fluorescence recovery is

$$\frac{N'(t)}{N} = \frac{\kappa(1-\kappa)T_f}{N}(1 - e^{-kt}) \equiv A_N(1 - e^{-kt}), \tag{10}$$

while the cytoplasmic recovery after nuclear bleach is given by

$$\frac{C'(t)}{C} = \frac{1}{C}\left(C_s + (1-\kappa)^2 T_f + \kappa(1-\kappa)T_f e^{-kt}\right) \equiv B_C + A_C e^{-kt}. \tag{11}$$

From fitting these functions to the measured FRAP curves we obtain $A_N, B_C, A_C$ and $k$. Because of the normalization two of these values are not independent: $A_C + B_C = 1$, so we are left with three measured values: $k$, $A_N$ and $A_C$, while we have four remaining unknown parameters: $N_s, C_s, k_{in}, k_{out}$. The nuclear to cytoplasmic ratio $r$ provides an additional measurement, but also introduces the additional unknown $\alpha$.

Although one might have thought bleaching the cytoplasm would provide additional information, it is a simple exercise to check that it does not. Specifically using $cb$ to label quantities after cytoplasmic bleaching, and defining

$$\frac{C_{cb}'(t)}{C} = A_C^{cb}(1 - e^{-kt}), \qquad \frac{N_{cb}'(t)}{N} = B_N^{cb} + A_N^{cb} e^{-kt}, \tag{12}$$

we find

$$A_C^{cb} = A_C, \qquad A_N^{cb} = A_N. \tag{13}$$

The conclusion is that we have one more parameter than we have measurements, leaving the model underdetermined. We resolve this by assuming $k_{in}$ does not depend on the signaling state of the cell, consistent with the literature (**Schmierer and Hill, 2005**), and hold it fixed as we solve for the other variables from the data using the equations

$$A_N = \frac{\kappa(1-\kappa)(1-N_s-C_s)}{N_s + \kappa(1-N_s-C_s)}, \qquad K = k_{in} + k_{out},$$

$$A_C = \frac{\kappa(1-\kappa)(1-N_s-C_s)}{C_s + (1-\kappa)(1-N_s-C_s)}, \qquad r = \alpha \frac{N_s + \kappa(1-N_s-C_s)}{C_s + (1-\kappa)(1-N_s-C_s)} \tag{14}$$

Although for any $k_{in}$ there is a solution to these equations, sensible solutions have $0 \leq N_s, C_s \leq 1$ and $k_{out} > 0$ and fortunately this requirement strongly constrains the value of $k_{in}$. With the corrections and fitting procedures discussed in the following sections we find that for sensible solutions we must have $6.6 \times 10^{-4} s^{-1} \leq k_{in} \leq 1.1 \times 10^{-3} s^{-1}$, and that within this range the qualitative behavior of the inferred parameters does not change. We therefore fixed $k_{in}$ in the middle of this range at $k_{in} = 8.9 \times 10^{-4} s^{-1}$.

## Practicalities

In practice it turns out to be difficult to bleach the nucleus perfectly and also to prevent the cytoplasm from bleaching somewhat. Due to fast diffusion within the cytoplasm and media, a uniform drop in cytoplasmic intensity can be observed at the time of nuclear photobleaching. Therefore, to represent the fraction of each compartment that is bleached, we define the nuclear and cytoplasmic bleach factors

$$b_n = \frac{I_n' - I_{bg}}{I_n - I_{bg}}, \qquad b_c = \frac{I_c' - I_{bg}}{I_c - I_{bg}} \tag{15}$$

Here $I$ and $I'$ denote the intensity right before and after photobleaching with subscript indicating the compartment. The background intensity is taken to be the mean intensity in areas without cells. We denote with a tilde, quantities that do not take this correction into account, and define their relationship with the original variables. Now the free fraction after bleaching is more complex

$$\tilde{T}_f' = ((1-\kappa)b_c + \kappa b_n)T_f. \tag{16}$$

Moreover, the nucleus starts with intensity $b_n T_f$ so the recovery amplitude is proportional to the difference between the new equilibrium and this initial value. The recovery curves become

$$\frac{\tilde{N}'(t)}{N} = \tilde{A}_{NN}(1 - e^{-kt}) + \tilde{B}_{NN} = \frac{\kappa(\tilde{T}_f' - b_n T_f)}{N}(1 - e^{-kt}) + b_n, \tag{17}$$

and

$$\frac{\tilde{C}'(t)}{C} = \tilde{A}_{NC} e^{-kt} + \tilde{B}_{NC} = \frac{(1-\kappa)(b_c T_f - T_f')}{C} e^{-kt} + \frac{(1-\kappa)T_f' + b_c C_s}{C}. \tag{18}$$

The fit parameters $A, B$ are not constrained to sum to one, with $B$ now containing additional information about the degree of bleaching for both compartments

$$\tilde{B}_{NN} = b_n, \qquad \tilde{A}_{NC} + \tilde{B}_{NC} = b_c. \tag{19}$$

The equations relating the measured amplitudes to the inferred parameters are now multiplied by a factor $b_c - b_n$,

$$A_{NN} \rightarrow \tilde{A}_{NN} = (b_c - b_n)A_{NN}, \qquad A_{NC} \rightarrow \tilde{A}_{NC} = (b_c - b_n)A_{NC}, \tag{20}$$

while the baseline for the cytoplasm transforms as

$$B_{NC} \rightarrow \tilde{B}_{NC} = (b_c - b_n)B_{NC} + b_n. \tag{21}$$

There appears to be no expression for nuclear to cytoplasmic ratio after recovery $\tilde{R}'$ in terms of only $R'$, but we can get the corrected $R'$ from plugging the corrected amplitudes ($A$, rather than the measured $\tilde{A}$) into (9).

## Parameter inference and sensitivity

To infer $N_S, C_s, k_{out}$ we used a weighted least square fit, which minimized

$$E(k_{in}, k_{out}, C_s, N_s) = \sum_i E_i^2, \quad E_i = \frac{p_i(p') - \bar{p}_i}{S_{p_i}}, \quad p_i(p') = p_i(k_{in}, k_{out}, C_s, N_s) \tag{22}$$

using the built-in MATLAB function lsqnonlin. Here $p$ denotes the measurements in (14), for example $p_1 = A_N$, while $p'$ denotes the inferred parameters. $\bar{p}$ denotes the mean of $p$, and $S_p$ the standard error in the mean of $p$. The values of the inferred parameters can be found in *Figure 1* of the main manuscript. The main feature is that adaptation is caused by changes in $N_s, C_s$ rather than $k_{in}, k_{out}$.

Error bars in the inferred parameters were determined from the error in the input parameters by error propagation. Specifically, if we write (22) in matrix notation, with $\delta p = p - \bar{p}$ and $S$ the diagonal matrix of standard errors, then

$$E = \delta p^T S^{-1} \delta p = \delta p'^T J^T S^{-1} J \delta p', \quad J_{ij} = \frac{\partial p_i}{\partial p'_j}, \tag{23}$$

where $J$ is the Jacobian, so the errors in the inferred parameters are given by the square root of the diagonal elements of

$$S' = (J^T S^{-1} J)^{-1}. \tag{24}$$

To understand the sensitivity of the measured quantities to changes in the inferred parameters, we can define a sensitivity matrix

$$N_{ij} = J_{ij} \frac{p'_j}{p_i} \tag{25}$$

which gives the relative change in the measured parameter given for a given relative change in the inferred parameter. Ordering the measurements (column) as $(A_N, A_C, k, R)$ and the parameters as $k_{out}, C_s, N_s$, for the parameter values in *Figure 1* the sensitivity matrix is given by

$$N_{ij}^{untreated} = \begin{pmatrix} -0.22 & -0.25 & -0.46 \\ -0.81 & -0.65 & -0.041 \\ 0.86 & 0 & 0 \\ -0.58 & -0.4 & 0.42 \end{pmatrix}, \quad N_{ij}^{peak} = \begin{pmatrix} ccc0.015 & -0.11 & -0.35 \\ -0.57 & -0.35 & -0.028 \\ 0.76 & 0 & 0 \\ -0.73 & -0.33 & 0.32 \end{pmatrix} \tag{26}$$

$$N_{ij}^{adapted} = \begin{pmatrix} 0.19 & -0.07 & -0.093 \\ -0.82 & -1.2 & -0.018 \\ 0.74 & 0 & 0 \\ -0.72 & -0.75 & 0.073 \end{pmatrix} \tag{27}$$

This effectively decomposes the error in the inferred parameters and for example tells us that $A_C$ mostly constrains $k_{out}$ and $C_s$, but is not sensitive to $N_s$.

## Mathematical modeling of adaptation mechanisms

In this section we extend the purely kinematic model of the previous section to a dynamic model; that is, we explore the predictions of different models for the mechanism of adaptation. In the section titled 'Receptor degradation cannot explain Activin/Nodal signaling

dynamics', we show that models where adaptive signaling is caused by depletion of upstream pathway components are not only inconsistent with our FRAP measurements, as noted in the main text, but are also inconsistent with the approximately equal timescales of adaptation and recovery that we observed in our pulse experiments. In contrast, we show in the section titled 'Feedback and feed-forward models are consistent with observed Activin/Nodal signaling dynamics', that either negative feedback or incoherent feed-forward models can account for all observed features of the signaling response. Finally, we show in the section titled 'Ligand degradation is sufficient to explain BMP dose response', that the BMP4 signaling response can be accounted for by ligand degradation.

## Minimal model

We consider models for signaling in which a signal transducer can be localized to either the nucleus or the cytoplasm. We assume that the production and degradation of the signal transducer are slow compared to the signaling dynamics so that the total amount of signal transducer is fixed. We adopt the notation from the simple model above ('Kinetic model'). Denoting the total amount of transducer by $T_f$ and the amount in the nucleus by $N_f$, we can write down a simple equation for the nuclear fraction as:

$$\dot{N}_f = k_{in}(I)(T_f - N_f) - k_{out}(I)N_f \tag{28}$$

where $k_{in}$ and $k_{out}$ are the rates of nuclear import and export, respectively, which can both depend on the concentration of an upstream component, $I$, that can be considered the ligand, the activated receptor, or the receptor-associated Smads. As discussed in the previous section, we choose $k_{in}$ to be independent of $I$ and assume that $k_{out}$ is reduced by $I$ with half-saturation constant $K$,

$$k_{in}(I) = k_{in}, \qquad k_{out}(I) = \frac{k_{out}^{(0)}}{K + I}. \tag{29}$$

At steady state

$$\bar{N}_f(I) = \frac{k_{in} T_f}{k}, \tag{30}$$

where $k$ is the sum of the in and out rates $k = k_{in} + k_{out}(I)$. If the input goes from $I_0$ to $I$, then the dynamics of $N_f$ are given by

$$N_f(t) = \bar{N}_f(I_0) + (1 - e^{-kt})(\bar{N}_f(I) - \bar{N}_f(I_0)). \tag{31}$$

That is, $N_f$ exponentially approaches its new steady state level with time scale $1/k$. As is to be expected, in this simple model, there is no adaptation.

## Receptor degradation cannot explain Activin/Nodal signaling dynamics

A simple mechanism of adaptation would be if the upstream component denoted by $I$ was degraded upon activation. To be concrete, we assume that $I$ represents activated receptors, $I_i$ represents inactivated receptors, and that receptors are activated by a ligand with concentration $L$, where we have subsumed the rate constant for activation into $L$ for simplicity. Then

$$\dot{I} = LI_i - (\gamma + \delta)I, \qquad \dot{I}_i = \beta - (\delta_i + L)I_i + \gamma I, \tag{32}$$

where $\beta$ represents production of receptors, $\gamma$ represents return of receptors to the inactive state, and $\delta, \delta_i$ respectively represent the degradation rates of activated and inactivated receptors. At steady state, we have

$$\bar{I} = \frac{\beta L}{\delta(L + \delta_i) + \gamma \delta_i}, \qquad \bar{I}_i = \frac{\beta(\gamma + \delta)}{\delta(L + \delta_i) + \gamma \delta_i}. \tag{33}$$

Note that when $\delta_i = 0$, $\bar{I} = \beta/\delta$ becomes independent of $L$ so that the model is perfectly adaptive in this case. However, in this case $\bar{I}_i = \beta(\gamma + \delta)/(\delta L)$ and so diverges as $L$ goes to 0.

At non-zero $\delta_i$, there is a ligand-dependent baseline that eventually saturates at the level $\beta/\delta$. We first solve the homogeneous equation

$$\dot{x} = Mx, \quad x = \begin{bmatrix} I' \\ I'_i \end{bmatrix}, \quad M = \begin{bmatrix} -(\delta + \gamma) & L \\ \gamma & -(L+\delta_i) \end{bmatrix}, \tag{34}$$

where $I' = I - \bar{I}$ and $I'_i = I_i - \bar{I}_i$. The solutions to the homogeneous equation can be written as

$$x(t) = Av_+ e^{\lambda_+ t} - Bv_- e^{\lambda_- t}, \tag{35}$$

where the eigenvalues are given by the solutions to the characteristic polynomial for $M$

$$\lambda_\pm = \frac{1}{2}[-(L+\gamma+\delta+\delta_i) \pm \sqrt{(L+\gamma+\delta+\delta_i)^2 - 4[\delta(L+\delta_i)+\gamma\delta_i]}], \tag{36}$$

and the (non-normalized) eigenvectors are

$$v_\pm = \begin{bmatrix} 1 \\ (\lambda_\pm + \gamma + \delta)/L \end{bmatrix}. \tag{37}$$

By taking the gradient of the term inside the square root in the eigenvalue equation (**Equation. 36**) in the space of the four parameters $L$, $\delta_i$, and $\gamma$, it is straightforward to show the minimum of this term is 0 and therefore it is always positive. The system is therefore over-damped and will always decay to the equilibrium value.

To proceed further, we consider the case $\gamma = 0$ which corresponds to receptor binding being irreversible. This is a reasonable assumption as off-rates for ligand-receptor binding are generally small (**Sako et al., 2010**), and further this is only way to achieve a timescale of adaptation that is independent of the ligand levels, consistent with our data. In this case, steady state levels reduce to

$$\bar{I} = \frac{\beta}{\delta}\frac{L}{L+\delta_i}, \quad \bar{I}_i = \frac{\beta}{L+\delta_i}, \tag{38}$$

and the eigenvalues to

$$\lambda_+ = -\delta; \quad \lambda_- = -(L+\delta_i). \tag{39}$$

The eigenvalue $\lambda_-$ will set the timescale for the signaling response following ligand exposure, and $\lambda_+$ will give the timescale to adapt, which depends only on the degradation constant for activated receptors and is therefore not dependent on the ligand concentration.

We now solve for the dynamics of the non-homogeneous system. We assume that initially $L = 0$, and the system is in equilibrium at this value. That is, $I(0) = \bar{I}(L = 0) = 0$ and $I_i(0) = \bar{I}_i(L = 0) = \beta/\delta_i$. These initial conditions yield the equations

$$0 = \bar{I} + A - B; \quad \frac{\beta}{\delta_i} = \bar{I}_i + \frac{L+\delta_i-\delta}{L}B. \tag{40}$$

Solving these for $A$ and $B$ yields

$$A = \frac{\beta L(\delta - \delta_i)}{\delta\delta_i(L+\delta_i-\delta)}, \quad B = \frac{\beta L^2}{\delta_i(L+\delta_i)(L+\delta_i-\delta)}. \tag{41}$$

In order for the system to adapt, the maximum value that $I$ reaches must be significantly higher than the baseline. Denoting the time at which this maximum is achieved by $t_{max}$, we have

$$\dot{I}(t_{max}) = A\lambda_+ e^{\lambda_+ t_{max}} - B\lambda_- e^{\lambda_- t_{max}} = 0, \tag{42}$$

which yields

$$t_{max} = (\delta - (L + \delta_i))^{-1} \ln \frac{\delta - \delta_i}{L}. \tag{43}$$

This implies that as $\delta_i \to \delta$, $t_{max} \to \infty$. This is logical because if the degradation rates of activated and inactive receptors are the same, there will be no adaptation and the signaling will reach its maximum at $t = \infty$. Adaptation therefore requires $\delta_i < \delta$.

Defining the maximum relative to baseline, $D$, as a measure of adaptation, and plugging in the values for $A$, $B$, and $\lambda_\pm$, we obtain after some algebra

$$D \equiv \frac{I(t_{max}) - \bar{I}}{\bar{I}} \tag{44}$$

Given that $0 < \delta_i < \delta$ and $t_{max} > 0$, we find that

$$D < \frac{\delta}{\delta_i}. \tag{45}$$

This implies that to explain the observed $D \approx 4$ (**Figure 1d**), we must have $\delta_i < \delta/4$. Now consider if the ligand is removed so that $L$ returns to 0. The system will regain the ability to respond to ligand following adaptation once the number of inactive receptors recovers. From **Equation. (39)**, the rate of recovery in this case is $\lambda_- = \delta_i$, and recall from the same equation that the rate of adaptation is $\delta$. With $\delta_i \ll \delta$ this means that the recovery from adaptation (refractory period) will take significantly longer than the time to adapt. This is not consistent with our data on Activin pulses in **Figure 5a,b** which shows that a full response can obtained with 5 hr between pulses, similar to the time required to adapt initially. This model is thus excluded as inconsistent with our data. As discussed in the main text and previous section it is also not consistent with our FRAP measurements which show that the adapted state is kinetically distinct from the pre-stimulation state. A model with degradation of upstream components would predict similar kinetics in these two cases.

## Feedback and feed-forward models are consistent with observed activin/nodal signaling dynamics

Feedback and feed-forward models provide alternative mechanisms of adaptation. In these models signaling activates an additional component which then causes inactivation of Smad4 and re-localization to the cytoplasm. This component could either be a Smad4 target (feedback model) or induced in parallel such as if it required only Smad2 or was induced by non-canonical signaling (feed-forward model). To be general, we extend the model from above ('Minimal model') to

$$\dot{x} = a \frac{L}{K + L} + bN_f - dx \quad \dot{N}_f = k_{in} - \frac{k_{out}}{K + L} N_f - cxN_f \tag{46}$$

The variable $N_f$ again represents the nuclear Smad4. The variable $x$ represents a component that inhibits Smad4 with the strength of this inhibition governed by the parameter $c$. This component can either be induced directly by the ligand $L$ (feed-forward) or by Smad4 (feedback), and the strengths of these inductions are controlled by the parameters $a$ and $b$, respectively. $x$ decays with rate $d$.

This model is not analytically tractable due to the non-linear term, so we examined whether we could reproduce the data presented in this paper for a variety of different combinations of $a$ and $b$. In particular, we considered whether we could accurately reproduce the dose response and the comparison between ramps and steps. **Appendix 1—figure 1** shows the dose response of Smad4 ($N_f$) to changing $L$. Although both the feed-forward and feedback models are capable of reproducing the features of the experimentally determined dose response, the feedback model does so more robustly. While many different values of the feedback strength ($b$) give the qualitatively correct behavior, when the feed-forward induction ($a$) is too strong, the curves tend to cross in the dose response due to a lowering of the steady-state value of $N_f$ with increasing ligand. Both models are also capable of reproducing

the comparison of the step versus the ramp (*Appendix 1—figure 2*) with the same caveat that high values of $a$ produced baselines that were reduced compared to the starting levels.

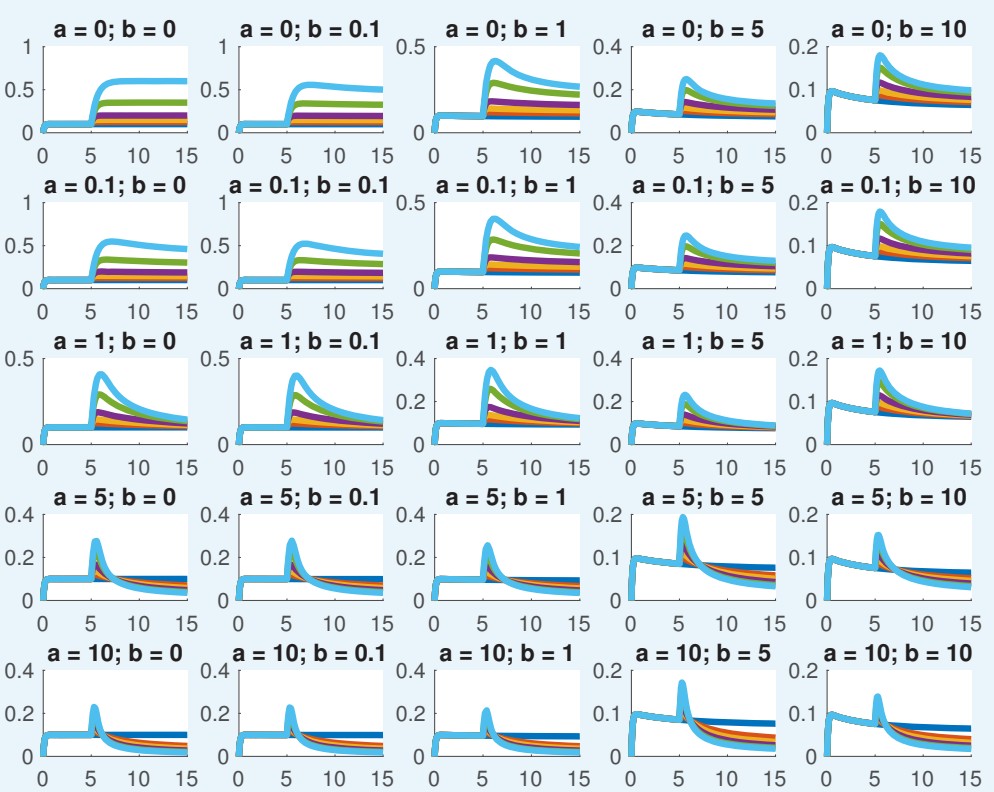

**Appendix 1—figure 1.** Simulated dose responses for the indicated values of $a$ and $b$. The x-axis indicates time and the y-axis nuclear Smad4, both are in arbitrary units. The values of $L$ used were 0, 0.5, 1, 2, 5, and 10. The values of the other parameters are $k_{in} = 1$, $k_{out} = 20$, $c = 1$, $d = 0.1$.

DOI: https://doi.org/10.7554/eLife.40526.027

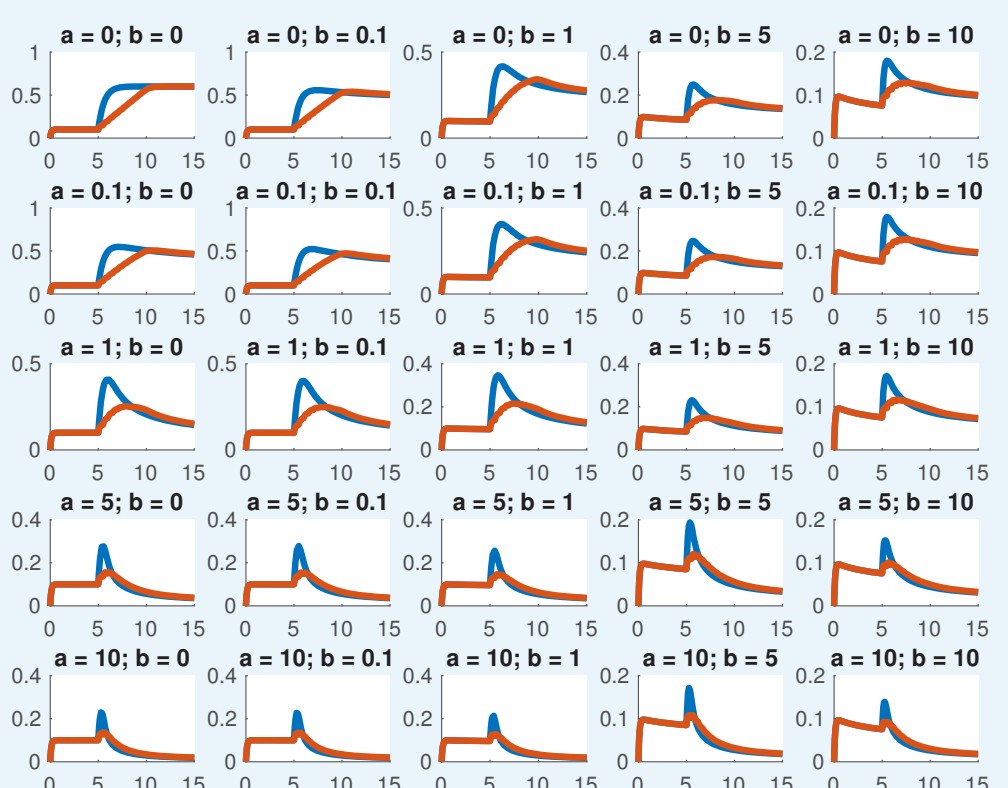

**Appendix 1—figure 2.** Simulated step and ramp responses for the indicated values of $a$ and $b$. Blue lines indicate a step increase in ligand to $L = 10$ while red lines indicate a ramp where the same level was achieved in 10 steps which were spaced by 0.5 time units each. The x-axis indicates time and the y-axis nuclear Smad4, both are in arbitrary units Other parameters are the same as in *Appendix 1—figure 1*.

DOI: https://doi.org/10.7554/eLife.40526.028

## Ligand degradation is sufficient to explain BMP dose response

Although the receptor degradation model is not consistent with Activin response, if irreversible receptor binding is taken to imply joint degradation of ligand/receptor complex in the presence of a finite amount of ligand, it can account for the observed dependence of Smad4 signaling on BMP4 dose. Without assuming irreversible binding, we can supplement *Equations (32)* with an equation accounting for the free ligand

$$\dot{L}(t) = -L(t)I_i(t) + \gamma I(t) \tag{47}$$

Integrating the model shows that duration of signaling depends on the initial ligand level, and larger supersaturating doses show sustained signaling for longer times (*Appendix 1—figure 3*), as experimentally observed for BMP4 (*Figure 1e*).

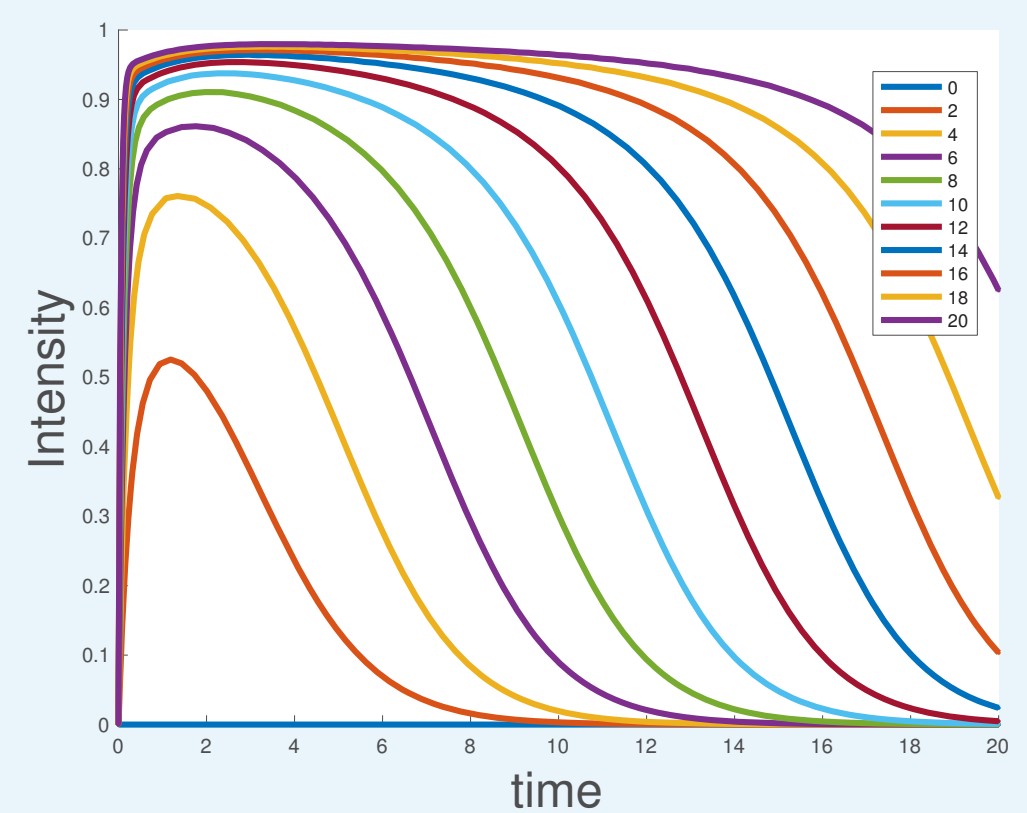

**Appendix 1—figure 3.** Simulated dose-response for the ligand depletion model, showing that this accounts for the BMP4 dose-response. The parameters used are $\gamma = \beta = \delta = 1$ and $\delta_i = 0.1$. The initial conditions were $I = 0$, $I_i = 1$ and $L = L_0$, and the value of $L_0$ was varied as indicated in the legend.

DOI: https://doi.org/10.7554/eLife.40526.029

