## [Decision Letter]

[**Editorial note:** This article has been through an editorial process in which the authors decide how to respond to the issues raised during peer review. The Reviewing Editor's assessment is that all the issues have been addressed.]

Acceptance notification:

The manuscript addresses the important question whether hESCs respond to changing concentrations of growth factors, in particular how BMP4 and Nodal control the various cell-fate decisions that hESCs take. It is shown that the response to BMP4 depends on the absolute concentration of the factor. In contrast, the response to the BMP-related Nodal signal depends on the rate of change in concentration. These findings will guide other investigators to consider the dynamics of signaling pathways and their concentrations in experiments manipulating stem cells differentiation.

Decision after peer review:

Thank you for submitting your article "Rapid changes in morphogen concentration control self-organized patterning in human embryonic stem cells" for consideration by *eLife*. Your article has been reviewed by three peer reviewers, and the evaluation has been overseen by a Reviewing Editor and Naama Barkai as the Senior Editor. The following individuals involved in review of your submission have agreed to reveal their identity: Kyle M Loh (Reviewer #1) and Roy Wollman (Reviewer #3). Reviewer #2 remains anonymous.

By measuring the dynamics of an endogenously tagged Smad4 fluorescent reporter, the authors provide evidence for adaptive responses of theTGF-β pathway to Nodal/Activin morphogens, They demonstrate that hESCs at sparse densities can sense the rate of change of ligand concentration. This work presents a quantitative approach to an important developmental pathway and should be of broad interest.

Major concerns:

The reviewers make a number of suggestions for further improvement of the manuscript. The comments are mostly restricted to the interpretation of the experiments and would require few if any new experimental data. We suggest taking these comments into account in resubmitting this work to *eLife*.

Separate reviews (please respond to each point):

*Reviewer #1:*

The temporal kinetics with which signaling pathways operate have generally been overlooked. Previous work by Warmflash and colleagues in cultured C2C12 cells suggested that TGFβ signaling is transient (i.e., cells respond to TGFβ ligands only for several hours) and adaptive (i.e., cells do not sense the absolute concentration of TGFβ ligands but rather they sense the rate of change, with large and sudden changes in TGFβ concentrations leading to the strongest responses) (Warmflash et al., 2012 and Sorre et al., 2014). However, whether C2C12 cells are a physiological model is questionable. In recent work with human embryonic stem cells (hESCs) it seems that a closely related signaling pathway, BMP, shows quite different dynamics: BMP4 treatment of hESCs leads to sustained (not transient) BMP signaling for many hours (Nemashkalo et al., 2017).

In the present submission, Heemskerk et al. perform a side-by-side comparison of the temporal dynamics of TGFβ and BMP signaling in hESCs. Monitoring nuclear translocation of SMAD4 by live imaging, they confirm that BMP4 treatment leads to sustained SMAD4 nuclear translocation in hESCs (thus reproducing their prior results reported in Nemashkalo et al., 2017) whereas ACTIVIN treatment leads to transient SMAD4 nuclear localization (Figure 1B-D). Indeed, many BMP target genes show sustained upregulation in response to BMP4, whereas TGFβ target genes show a mixed pattern of either transient or sustained upregulation in response to ACTIVIN (Figure 2). Using sophisticated microfluidic methods to control the rate of ligand delivery, they suggest that TGFβ (but not BMP) signaling relies on the change of ligand concentration (i.e., its time-derivative but necessarily the absolute concentration; Figure 3) and further that there is a traveling wave of TGFβ activity in BMP4-treated hESC colonies of defined sizes (Figure 4). By virtue of the transient and adaptive nature of TGFβ signaling, the authors conclude by suggesting that temporal pulses of ACTIVIN treatment might be best to efficiently upregulate BRACHYURY (Figure 4D).

Overall, the suggestion that TGFβ and BMP signaling kinetics (on the scale of hours) could be quite different is intriguing. The Discussion is excellently written, and appropriately emphasizes that a mechanistic understanding of the kinetics with which these pathways signal could greatly enrich both stem cell and developmental biology.

Major critiques:

1) SMAD4. The authors measure the activity of the TGFβ as well as BMP pathways by principally assessing a single metric: SMAD4 nuclear localization. However it is controversial whether SMAD4 localization as a single metric can accurately describe the total levels of TGFβ and BMP pathway activity. There are some reports that only a subset of TGFβ target genes require SMAD4 (e.g., Levy and Hill, 2005). If there is SMAD4-independent TGFβ signaling, it is therefore plausible that certain TGFβ target genes could still be transcribed (i.e., the TGFβ pathway is activated) in the window of time when SMAD4 is not in the nucleus (i.e., in the "adaptation" phase that the authors identify). If so, SMAD4 nuclear localization might only partially readout TGFβ signaling, which could therefore complicate the interpretation of the authors' data. Indeed, the authors show that upon ACTIVIN exposure for 5 hours (a timepoint by which SMAD4 has largely vacated the nucleus; Figure 1D), a number of TGFβ target genes (LEFTY1, NODAL, CER1 and WNT3) still continue to be transcribed (Figure 2D), indicating the possibility that SMAD4 nuclear localization does not completely reflect the degree/extent of TGFβ signaling. Could the authors perform luciferase reporter assays of ACTIVIN-treated hESCs to examine whether TGFβ transcriptional responses are truly transient? It would be preferable for the authors to perform live imaging of endogenously-expressed, labeled SMAD2/3 or SMAD1/5/8 proteins to examine their subcellular localization upon ACTIVIN or BMP treatment, respectively, but this is unreasonable to ask the authors to conduct in a short span of time. Barring those experiments, the authors might want to confine their conclusions to refer to "SMAD4-dependent TGFβ and BMP signaling" as opposed to "TGFβ and BMP signaling". For instance, would it be more accurate to specifically say that SMAD4-dependent TGFβ signaling (as opposed to TGFβ signaling altogether) is transient and adaptive?

2) SMAD4 signal-to-noise ratio. Their live imaging data suggests that upon ligand stimulation, the increase in nuclear SMAD4 is modest (2-fold or less; Figure 1D). This is rather surprising, because one might anticipate a massive change in SMAD4 levels is needed to drive large transcriptional changes. The modest SMAD4 signal-to-noise ratio also raises questions about the authors' subsequent and quite detailed analysis of temporal dynamics, adaptation and so on. Why is the signal-to-noise ratio so modest? One possibility is that there are high nuclear SMAD4 levels in the steady-state (i.e., there is high SMAD4 background), perhaps because the authors' hESCs are grown in mTeSR1 (Materials and methods), which contains 2 ng/mL of TGFβ1 (Ludwig et al., 2006; Nature Cell Biology), and would be expected to lead to some level of basal nuclear SMAD4. What if the authors transiently pulsed their hESCs with a TGFβ inhibitor before performing the various ACTIVIN and BMP stimulations, would this reduce their background and thus enhance the signal-to-noise ratio?

Minor critiques:

1) Figure 1H-J should be more clearly labeled and explained.

2) "Error bars" erroneously merged into 1 word (Figure Legends).

3) Data in the Nemashkalo et al., 2017, paper (by the same authors) seems to already suggest that BMP4 leads to sustained SMAD4 signaling, therefore it seems appropriate for Heemskerk et al. to cite this earlier work, as their Figure 1 seems to be confirming-and extending beyond-this earlier paper.

4) There seem to be some strange artifacts in certain subpanels. For instance why does the orange line in Figure 3B (reflecting nuclear SMAD4 localization after sudden exposure to high BMP) jitter at regular intervals? It makes sense that the blue lines in Figure 3B,E would jitter (as they might correlate with each step of the ligand concentration increase) but not the orange line in Figure 3B.

5) Similarly, in Figure 4B (red line: pulses), there seem to be discontinuous spikes in SMAD4 whenever the media is changed.

6) The authors conclude that BMP4 induces sustained, whereas ACTIVIN induces transient, SMAD4 signaling, yet in Figure 4 they suggest that BMP4 might induce endogenous TGFβ signaling. One alternate interpretation of the authors' data is that maybe BMP4 itself induces a transient SMAD4 response, but that by secondarily inducing TGFβ signaling, this latter wave of TGFβ might lead to sustained SMAD4.

7) Another oddity is in Figure 3C. If hESCs sense the absolute concentration of BMP4 ligand, one would expect that at low levels of BMP, they would respond weakly and at higher levels of BMP, they would respond strongly. Yet in Figure 3C, at the 2-3 hr timepoint, the "ramp" cells (blue line, exposed to low BMP4 at that point in time) express HAND1 mRNA almost as highly as the "step" cells (orange line, exposed to high BMP4 at that point in time). The only other interpretation I can think of is that maybe the low BMP4 levels in the blue line (at 2-3 hrs) are super-threshold already.

*Reviewer #2:*

Heemskerk et al. provide evidence for adaptive responses of the TGF-β pathway to Nodal/Activin morphogens. The authors elegantly measured dynamics of an endogenously tagged Smad4 fluorescent reporter that transduces both BMP and Nodal/Activin input signals, and demonstrate that at sparse densities hESCs can sense the rate of change of ligand concentration. Using mathematical models and experimental measurements of endogenously tagged Smad4, they rule out some mechanisms of pathway adaptation, and propose that Nodal/Activin adaptation occurs through reduced nuclear sequestration and/or increased cytoplasmic sequestration. Under sparse growth conditions, they also find that endogenous target genes can selectively respond to pulsatile and sustained modes of Smad4 activation resulting from Activin and BMP inputs, respectively. The authors suggest that sensing the rate of change of ligand concentration, demonstrated in sparse cultures, could be used by the mammalian embryo during mesodermal patterning, and use an in vitro model of human gastrulation to test this hypothesis. In this system, they correlate a moving front of Smad4 nuclear localization with the timing and spatial position of Brachyury expression, a transcription factor involved in mesodermal differentiation.

Overall, this study brings a quantitative perspective to a central developmental pathway, and should help to further strengthen a growing sense that dynamics are critically important in the behavior of many developmental signaling pathways. It suggests that controlling ligand dynamics could facilitate directed differentiation protocols. It should therefore be of broad interest, and we support publication.

Below, we describe several issues that could be addressed to strengthen the work in a revised manuscript:

Major comments:

Figures 1 and Figure 1—figure supplement 1F together show that BMP4 triggers a more stable nuclear localization of BMP-responsive Smad1, while activin results in adaptive nuclear localization of Nodal-responsive Smad2/3. However, the cross-pathway dynamics are unclear. Does BMP4 affect Smad2/3 and does Activin/Nodal affect Smad1, and if so, with what dynamics? This control is important to support the authors claim that the difference in dynamics results from distinct pathway-specific transducers.

Figure 2, B and D, present gene sets that show qualitatively different dynamic response to Activin. The authors suggest the interesting possibility that these sets have different sensitivities to Smad4. However, an alternative explanation could be a difference in RNA half-lives, with the monotonic responses in Figure 2D resulting from stable mRNAs that accumulate over time, effectively integrating the response. Is it possible to discriminate between these explanations (e.g. by examining nascent or intronic RNA), or at least discuss these two possibilities?

Figure 3 presents an elegant comparison between ramps (staircases) and steps of BMP and Activin. The results are striking. However, two issues complicate interpretation. First, the ramp and step differ in total integrated signaling over time, making it difficult, from this single experiment, to disentangle the roles of ligand concentration and dynamics. The simplest way to address this might be to also include a step of half the amplitude, or to do a ramp at double the slope. Second, based on Figure 1 and Figure 1—figure supplement 1B-C, which shows dose-dependent responses to BMP concentration, we would expect the ramp response in Figure 3B to depend on ligand dose and therefore to grow slowly over time. Why does it step up so rapidly?

In Figure 3, rate responsive behavior is demonstrated by comparing a step to a ramp. Does the Nodal pathway act as a quantitative sensor of the rate of increase of Activin, or does it exhibit some kind of rate threshold? This could be answered by showing how response depends quantitatively on the ramp slope. A mathematical model of the adaptation process could help to provide insight into the expected behavior here.

Rate-responsiveness in other systems. Work in bacteria (Young et al., 2013) showed similar rate-responsive behavior in the sigB stress response system, where two kinds of stresses activate sigB in different ways, with one input showing rate-responsive behavior, assayed by temporal "ramps" of different slopes. In that system the mechanism of rate responsiveness has been determined. It would be interesting to discuss the similarity (and differences) in signal processing behaviors between these very different biological systems.

In Figure 4, the authors suggest that during development the moving Smad4 signaling front is responsible for the activation of Brachyury expression. Although strong evidence for adaptative and rate sensing responses to Nodal/ActivinA ligands is provided for sparse hESCs cultures, the micropattern experiment cannot distinguish between a rate sensing mechanism and a pure concentration sensing mechanism. Even though Brachyury responds with higher expression to Activin pulses in sparse hESC cultures, these two experimental conditions differ in both the time scales and the number of pulses cells could perceive (i.e. in sparse culture cells are exposed to multiple Activin pulses separated by ~4.5 hours, whereas in micropattern the data is consistent with only one wave of pSmad2/3, one wave of pSmad1 and one front of Smad4 moving at 1 cell diameter (10µm) per hour, and traveling at least 150 µm to fully cover the spatial domain of brachyury expression. Since no data on dynamics of endogenous Nodal expression is provided, it remains unknown if, over time, ligand concentration is locally increasing (or oscillating) on a timescale relevant to Brachyury induction. As result, in the micropattern, in which responses to BMP and endogenous Nodal co-occur, it is unclear if cells are sensing absolute ligand concentration and/or a rapid increase in ligand concentration. Experimentally determining Nodal dynamics in micropatterns remains difficult, and perturbing its dynamics would require significant cell line engineering. Thus, an alternative avenue to support the claim that in the gastrulating human embryo expression of fate determinants results from rate sensing of ligand increments could be to computationally compare whether patterning triggered by adaptation to Nodal achieved by rate sensing rate qualitatively or quantitatively differs from patterning the mesoderm through sensing absolute ligand concentrations. What new or different emergent properties result from sensing rates as opposed to just concentrations?

Relation to other dynamic encoding systems: This study provides an important advance in understanding the role of dynamics in core mammalian signaling systems. But the relationship to other work on dynamic signal encoding in signaling pathways, as reviewed for example in Purvis and Lahav (2013), Behar and Hoffmann (2010), etc. is minimal. Specifically, it would be useful to know which aspects of the Nodal system in hES resemble or differ from phenomena observed in other pathways or other cell types. For example, different target programs are activated by sustained vs pulsatile activation of p53 and Notch, among other pathways. Also, a recent report (Frick et al., 2017 PNAS) studied the response of C2C12 cells to TGFb ligands, which belong to the Activin/Nodal family and are also transduced by Smads2/3, and showed that the pathway senses fold changes. How does this fold change detection relate to the adaptive rate-responsive behavior observed here? Discussion of these issues would help to connect this study to the broader literature and unify results from multiple studies.

Minor Comments:

In Figure 2, please include the base of the log on the y axis: is it 2, e, or something else?

In Figure 1E there is an orange line near the y-axis. It remains unclear if this line represents the fact that after photobleaching Smad4 fluorescence does not recover to the original amplitude.

*Reviewer #3:*

The paper by Idse et al. includes fantastic work by the authors to explore fundamental properties of mammalian signaling in an interesting developmental context using an innovative experimental system.

I am very happy to participate in this "peer-review trial in which the authors decide how to respond to the peer-review comments." and to learn that "the article will almost certainly be published".

I do not have any additional experiment to suggest, the authors did a good job providing the adequate controls and I have little doubt in the correctness of the results they present.

Perhaps the biggest suggestion I can make with regards to the work is that it is a bit too dense and hard to grasp. Overall the work follows three parts: 1) An analysis of mechanism for adaptation using innovative combination of FRAP and parameter identification of ODE model. 2) analysis of gene expression dynamics response to dynamic SMAD4, and 3) implication of the differences in response dynamics on overall self-organization patterns in the in-vitro gastrulation model. I will refer to these as parts 1-3 below for brevity.

I am not sure why the authors restricted themselves to only four figures, although I can make an educated guess;). To make the paper easier to follow I strongly recommend to take advantage of *eLife* lack of restriction on length and number of figures and use the space to elaborate much more about the work.

In part 1 the author only use two paragraphs on main text to explain a lot of stuff. To understand the adaptation and the FRAP experiments one has to think about two different math models that use a parameter fitting procedure to identify underlying mechanisms. The material is mostly in the supp text, but supp text should not replace main text rather provide more details for completeness. I recommend to move some of the material from supp mat to the main text, explain the FRAP procedure, show the alternative fits for different parameters etc. This part can easily take 2-3 figures.

Similarly, part II seemed rushed. The authors show how some genes follow SMAD4 dynamics and some do not. But the sparse text doesn't allow them to go into details, what is different about these genes? Is it RNA halflife? Even without additional experiments, few simple models would be very helpful to understand Figure 2A-D. I am not sure I got the point of panels 2EF, so either explain that in more details or remove those. The different gene expression response to different stimuli regimes (Figure 3) is very nice with its demonstration of the role of SMAD4 dynamics.

Finally, the last part is very short and descriptive, The author measure different SMADs over space and time, show that there are different responses and that these change gene expression. I was left a bit confused as to why these differences arise? What aspect of self-organization is in play to cause these patterns. The fact that all these results are described in a single paragraphs of 322 words might be part of the problem.

Minor Comments:

1) The use of SMAD4 signaling in figure titles should be replaced with y-axis label saying SMAD4_{nuc}

2) The cartoon in Figure 2I is confusing, why don't all the no sequestered dots recover?

---

## [Author Response]

Reviewer #1:

[…] Overall, the suggestion that TGFβ and BMP signaling kinetics (on the scale of hours) could be quite different is intriguing. The Discussion is excellently written, and appropriately emphasizes that a mechanistic understanding of the kinetics with which these pathways signal could greatly enrich both stem cell and developmental biology.Major critiques:1) SMAD4. The authors measure the activity of the TGFβ as well as BMP pathways by principally assessing a single metric: SMAD4 nuclear localization. However it is controversial whether SMAD4 localization as a single metric can accurately describe the total levels of TGFβ and BMP pathway activity. There are some reports that only a subset of TGFβ target genes require SMAD4 (e.g., Levy and Hill, 2005). If there is SMAD4-independent TGFβ signaling, it is therefore plausible that certain TGFβ target genes could still be transcribed (i.e., the TGFβ pathway is activated) in the window of time when SMAD4 is not in the nucleus (i.e., in the "adaptation" phase that the authors identify). If so, SMAD4 nuclear localization might only partially readout TGFβ signaling, which could therefore complicate the interpretation of the authors' data. Indeed, the authors show that upon ACTIVIN exposure for 5 hours (a timepoint by which SMAD4 has largely vacated the nucleus; Figure 1D), a number of TGFβ target genes (LEFTY1, NODAL, CER1 and WNT3) still continue to be transcribed (Figure 2D), indicating the possibility that SMAD4 nuclear localization does not completely reflect the degree/extent of TGFβ signaling. Could the authors perform luciferase reporter assays of ACTIVIN-treated hESCs to examine whether TGFβ transcriptional responses are truly transient?

We thank the reviewer for this suggestion. We have attempted to perform luciferase assays using two different TGFb responsive promoters described in the literature (3TP and CAGA), however, neither showed a strong response to Activin treatment in hESCs, indicating that there are likely cell-type specific cofactors necessary for these promoters. In agreement with this, we found by qPCR that the PAI1 gene, from which the 3TP promoter was derived, is also not induced by Activin treatment in hESCs. We agree with the reviewer that this is an important issue. We believe there are two possible, not mutually exclusive, interpretations of the gene expression data. (1) Most transcription is Smad4 dependent but low levels of Smad4 are sufficient for some genes. These genes are maintained by the baseline in Smad4 following adaptation and yield sustained transcription. (2) Only a subset of genes are Smad4 dependent, while others can be induced by Smad2 alone. Since Smad2 is more sustained, these genes are sustained. Of course, one of the explanations may be true for some genes and the other for other genes. We attempted to distinguish between these possibilities by using CRISPR/Cas9 to knock out Smad4 but we found that we were unable to maintain pluripotent Smad4-KO cells. We believe that the dataset presented accurately captures the fact that there are multiple different kinetic behaviors of downstream target genes. We have expanded our discussion to better reflect the different possible explanations.

It would be preferable for the authors to perform live imaging of endogenously-expressed, labeled SMAD2/3 or SMAD1/5/8 proteins to examine their subcellular localization upon ACTIVIN or BMP treatment, respectively, but this is unreasonable to ask the authors to conduct in a short span of time. Barring those experiments, the authors might want to confine their conclusions to refer to "SMAD4-dependent TGFβ and BMP signaling" as opposed to "TGFβ and BMP signaling". For instance, would it be more accurate to specifically say that SMAD4-dependent TGFβ signaling (as opposed to TGFβ signaling altogether) is transient and adaptive?

We agree with the reviewer that it would be of interest to perform live cell imaging with Smad1 and Smad2, but that these experiments are beyond the scope of the present paper. We note that immunofluorescence data for the activity of these signal transducers are presented in the supplementary data. As suggested, we also now explicitly clarify at the beginning of the results that we are referring to SMAD4-dependent TGFb and BMP signaling, but as this language is cumbersome, we use the shorthand “TGFb and BMP signaling” throughout. Finally, we believe that the discussion we incorporated on the dynamics of Smad2 vs Smad4 (see previous answer) also highlights this important point.

2) SMAD4 signal-to-noise ratio. Their live imaging data suggests that upon ligand stimulation, the increase in nuclear SMAD4 is modest (2-fold or less; Figure 1D). This is rather surprising, because one might anticipate a massive change in SMAD4 levels is needed to drive large transcriptional changes. The modest SMAD4 signal-to-noise ratio also raises questions about the authors' subsequent and quite detailed analysis of temporal dynamics, adaptation and so on. Why is the signal-to-noise ratio so modest? One possibility is that there are high nuclear SMAD4 levels in the steady-state (i.e., there is high SMAD4 background), perhaps because the authors' hESCs are grown in mTeSR1 (Materials and methods), which contains 2 ng/mL of TGFβ1 (Ludwig et al., 2006; Nature Cell Biology), and would be expected to lead to some level of basal nuclear SMAD4. What if the authors transiently pulsed their hESCs with a TGFβ inhibitor before performing the various ACTIVIN and BMP stimulations, would this reduce their background and thus enhance the signal-to-noise ratio?

We agree with the reviewer that the measured changes in Smad4 are likely smaller than the increase in the active Smad complexes which mediate transcription. Background stimulation from the TGFb1 in mTeSR1 does not explain the small change as adding the Activin inhibitor reduced the baseline Smad2 levels but not those of Smad4. This can be seen in Figure 1—figure supplement 1G, where the data are scaled so that the levels with SB are 1. The higher levels of SMAD2 compared to SMAD4 prior to Activin treatment show that SB reduces these baseline levels. We believe that the relatively small increase in nuclear Smad4 results from the fact that Smads are present in the nucleus even in the absence of stimulation, as under all conditions Smads shuttle between the nucleus and cytoplasm. So that even when the pathway is inactive, there is some Smad in the nucleus (see Schmierer, 2005). Also, background fluorescence which is present in both the nucleus and the cytoplasm will tend to bring the ratios closer to 1 both before and after stimulation, resulting in reduced fold changes. In any event, the nuclear to cytoplasmic ratio will be a monotonically increasing function of the total Smad4 in the nucleus, and so is a suitable reporter for Smad levels. We chose to use this ratio instead of absolute levels as it tends to remove noise associated with variations in Smad expression, cell size, and imaging artifacts, and we have found empirically that it is a more reliable and reproducible metric for signaling. We point out that the peak to baseline signaling ratio is not the same thing as signal to noise, which would be either of those levels relative to the standard deviation around that level. The significance of the difference between the two signaling states would be best captured by a statistical test such as Kolmogorov-Smirnov and the differences before and after stimulation are highly significant.

Minor critiques:1) Figure 1H-J should be more clearly labeled and explained.

We have substantially expanded the discussion of the FRAP experiments and split these off into a separate figure in order to clarify.

2) "Error bars" erroneously merged into 1 word (Figure Legends).

This has been corrected.

3) Data in the Nemashkalo et al., 2017, paper (by the same authors) seems to already suggest that BMP4 leads to sustained SMAD4 signaling, therefore it seems appropriate for Heemskerk et al. to cite this earlier work, as their Figure 1 seems to be confirming-and extending beyond-this earlier paper.

We have added this citation.

4) There seem to be some strange artifacts in certain subpanels. For instance why does the orange line in Figure 3B (reflecting nuclear SMAD4 localization after sudden exposure to high BMP) jitter at regular intervals? It makes sense that the blue lines in Figure 3B,E would jitter (as they might correlate with each step of the ligand concentration increase) but not the orange line in Figure 3B.

We believe that these artifacts result from slight perturbations to the imaging due to the media replacement. Although we could remove these by smoothing, we prefer to present the data as is without manipulation. We have added a note to figure caption explaining these artifacts to avoid confusion.

5) Similarly, in Figure 4B (red line: pulses), there seem to be discontinuous spikes in SMAD4 whenever the media is changed.

See answer to point 4.

6) The authors conclude that BMP4 induces sustained, whereas ACTIVIN induces transient, SMAD4 signaling, yet in Figure 4 they suggest that BMP4 might induce endogenous TGFβ signaling. One alternate interpretation of the authors' data is that maybe BMP4 itself induces a transient SMAD4 response, but that by secondarily inducing TGFβ signaling, this latter wave of TGFβ might lead to sustained SMAD4.

To exclude this possibility we have examined treatment with BMP together with a TGFb inhibitor and Activin together with a BMP inhibitor. These data support our conclusion that BMP signaling is sustained while Activin signaling is transient (Figure 1G,H).

7) Another oddity is in Figure 3C. If hESCs sense the absolute concentration of BMP4 ligand, one would expect that at low levels of BMP, they would respond weakly and at higher levels of BMP, they would respond strongly. Yet in Figure 3C, at the 2-3 hr timepoint, the "ramp" cells (blue line, exposed to low BMP4 at that point in time) express HAND1 mRNA almost as highly as the "step" cells (orange line, exposed to high BMP4 at that point in time). The only other interpretation I can think of is that maybe the low BMP4 levels in the blue line (at 2-3 hrs) are super-threshold already.

We thank the reviewer for pointing this out, and we discovered an error in our previous data – the earlier ramp was performed in steps of 1 ng/ml to 10 ng/ml, whereas we intended to use data from a ramp to 3 ng/ml in steps of 0.3 ng/ml. This slower ramp does show a slightly slower increase, particularly in the SMAD4 signaling. Nonetheless, the reviewer is correct that the ramp reaches the levels of the step quickly. The explanation for this is that the BMP dose response (now in Figure 1E) shows that there is a relatively sharp threshold in BMP signaling over which full activation is achieved. This agrees with our previous data (Nemashkalo et al., 2017), as well as from other labs who have suggested that BMP responses are close to binary (Etoc et al. Dev Cell 2016). Thus, it only takes a few steps in the ramp to reach full activation of the pathway. We note that the dose of BMP will have a graded effect on integrated signaling due to the decay in time of signaling at low doses (Figure 1E). We have revised our discussion of this experiment to reflect this, and of the BMP dose-response to better highlight the sharp transition between off and on.

Reviewer #2:

[…] Overall, this study brings a quantitative perspective to a central developmental pathway, and should help to further strengthen a growing sense that dynamics are critically important in the behavior of many developmental signaling pathways. It suggests that controlling ligand dynamics could facilitate directed differentiation protocols. It should therefore be of broad interest, and we support publication.Below, we describe several issues that could be addressed to strengthen the work in a revised manuscript:Major comments:Figures 1 and Figure 1—figure supplement 1F together show that BMP4 triggers a more stable nuclear localization of BMP-responsive Smad1, while activin results in adaptive nuclear localization of Nodal-responsive Smad2/3. However, the cross-pathway dynamics are unclear. Does BMP4 affect Smad2/3 and does Activin/Nodal affect Smad1, and if so, with what dynamics? This control is important to support the authors claim that the difference in dynamics results from distinct pathway-specific transducers.

We thank the reviewer for this suggestion. To address this issue, we have performed additional experiments in which each pathway is stimulated while the other is inhibited. The results show that BMP has no effect on the dynamics downstream of Activin while Activin/Nodal signaling has a small negative effect on the response to BMP so that inhibiting Activin during BMP signaling gives a stronger response. These data are now included as Figure 1G,H.

Figure 2, B and D, present gene sets that show qualitatively different dynamic response to Activin. The authors suggest the interesting possibility that these sets have different sensitivities to Smad4. However, an alternative explanation could be a difference in RNA half-lives, with the monotonic responses in Figure 2D resulting from stable mRNAs that accumulate over time, effectively integrating the response. Is it possible to discriminate between these explanations (e.g. by examining nascent or intronic RNA), or at least discuss these two possibilities?

To examine this possibility, we have performed gene expression experiments in which we inhibited Activin/Nodal signaling and examined target gene expression in time. If the differences in gene expression dynamics were due to differences in mRNA half-life, we would expect that transcripts that are stably induced in response to Activin stimulation would persist following pathway inhibition. However, we find that targets including Lefty, Nodal, and Nanog which are activated in a sustained manner rapidly decline upon inhibition of the pathway with the small molecule inhibitor SB431542. Thus, different mRNA stabilities cannot explain our gene expression results. This data is now included in Figure 3—figure supplement 1G and discussed in the text. In addition, while not impossible, it would be surprising if the different transcriptional dynamics of HAND1 in response to BMP4 vs Activin in Figure 3C and of Noggin in the supplementary data for Figure 3 were due to differential modulation of RNA life time as it is the same transcript being induced. The more natural interpretation is that this directly reflects SMAD4 dependent transcriptional activity.

Figure 3 presents an elegant comparison between ramps (staircases) and steps of BMP and Activin. The results are striking. However, two issues complicate interpretation. First, the ramp and step differ in total integrated signaling over time, making it difficult, from this single experiment, to disentangle the roles of ligand concentration and dynamics. The simplest way to address this might be to also include a step of half the amplitude, or to do a ramp at double the slope.

We now include data on a faster ramp, as suggested, and find that the response is still strongly reduced from step stimulation, however it is stronger than the slower ramp. These data support our conclusion that the time-derivative of concentration is the key determinant of the response rather than the integrated signal and are now presented in Figure 4G,H. We also point out that signaling responses are coincident by the end of the experiment, so that integrated signaling response to the ramp will never catch up with integrated signaling response to step, even if it were allowed to run longer to get to the same integrated ligand exposure.

Second, based on Figure 1 and Figure 1—figure supplement 1B-C, which shows dose-dependent responses to BMP concentration, we would expect the ramp response in Figure 3B to depend on ligand dose and therefore to grow slowly over time. Why does it step up so rapidly?

The BMP dose response (now in Figure 1E) shows that there is a relatively sharp threshold in BMP signaling over which full activation is achieved. This agrees with our previous data (Nemashkalo et al., 2017), as well as from other labs who have suggested that BMP responses are close to binary (Etoc et al., Dev Cell 2016). Thus, it only takes a few steps in the ramp to reach full activation of the pathway. We have revised our discussion of this experiment to reflect this, and of the BMP dose-response to better highlight the sharp transition between off and on.

In Figure 3, rate responsive behavior is demonstrated by comparing a step to a ramp. Does the Nodal pathway act as a quantitative sensor of the rate of increase of Activin, or does it exhibit some kind of rate threshold? This could be answered by showing how response depends quantitatively on the ramp slope. A mathematical model of the adaptation process could help to provide insight into the expected behavior here.

We now include data showing intermediate rates of increase which yield intermediate responses in the Smad4 signaling (Figure 4G,H). We also note that the supplement includes a model of the adaptation process that generates behavior consistent with these experiments (Supplemental text).

Rate-responsiveness in other systems. Work in bacteria (Young et al., 2013) showed similar rate-responsive behavior in the sigB stress response system, where two kinds of stresses activate sigB in different ways, with one input showing rate-responsive behavior, assayed by temporal "ramps" of different slopes. In that system the mechanism of rate responsiveness has been determined. It would be interesting to discuss the similarity (and differences) in signal processing behaviors between these very different biological systems.

We now cite this paper in the discussion, however, as our focus is on adaptation in developmental signaling, we feel a detailed discussion of this paper is beyond the scope of our manuscript.

In Figure 4, the authors suggest that during development the moving Smad4 signaling front is responsible for the activation of Brachyury expression. Although strong evidence for adaptative and rate sensing responses to Nodal/ActivinA ligands is provided for sparse hESCs cultures, the micropattern experiment cannot distinguish between a rate sensing mechanism and a pure concentration sensing mechanism. Even though Brachyury responds with higher expression to Activin pulses in sparse hESC cultures, these two experimental conditions differ in both the time scales and the number of pulses cells could perceive (i.e. in sparse culture cells are exposed to multiple Activin pulses separated by ~4.5 hours, whereas in micropattern the data is consistent with only one wave of pSmad2/3, one wave of pSmad1 and one front of Smad4 moving at 1 cell diameter (10µm) per hour, and traveling at least 150 µm to fully cover the spatial domain of brachyury expression. Since no data on dynamics of endogenous Nodal expression is provided, it remains unknown if, over time, ligand concentration is locally increasing (or oscillating) on a timescale relevant to Brachyury induction. As result, in the micropattern, in which responses to BMP and endogenous Nodal co-occur, it is unclear if cells are sensing absolute ligand concentration and/or a rapid increase in ligand concentration. Experimentally determining Nodal dynamics in micropatterns remains difficult, and perturbing its dynamics would require significant cell line engineering. Thus, an alternative avenue to support the claim that in the gastrulating human embryo expression of fate determinants results from rate sensing of ligand increments could be to computationally compare whether patterning triggered by adaptation to Nodal achieved by rate sensing rate qualitatively or quantitatively differs from patterning the mesoderm through sensing absolute ligand concentrations. What new or different emergent properties result from sensing rates as opposed to just concentrations?

While we agree in principle that these data in isolation are also consistent with sensing ligand concentration, we note that this experiment does provide an important test, and could have resulted in data disproving our model. The most straightforward hypothesis would be that a static gradient of Nodal signaling is formed, peaks in the region of future mesendoderm, and cells read this gradient. Had this been true, it would have shown that the rate of increase is not relevant to the patterning that occurs. Our data disproved this hypothesis. The fact that there is a rapidly moving wave of signal activation in micropatterned colonies together with our experiments in Figures 1 – 3 (now 1 – 5) that demonstrate that cells respond to the rate of Activin/Nodal increase strongly suggest that cells in micropatterning colonies are also responding to the rate of increase. We agree that direct demonstration would require measurement of Nodal protein dynamics but believe this to be beyond the scope of this manuscript. We also note that the consequences of reading rate of increase during patterning were modeled theoretically in Sorre et al., 2014.

Relation to other dynamic encoding systems: This study provides an important advance in understanding the role of dynamics in core mammalian signaling systems. But the relationship to other work on dynamic signal encoding in signaling pathways, as reviewed for example in Purvis and Lahav (2013), Behar and Hoffmann (2010), etc. is minimal. Specifically, it would be useful to know which aspects of the Nodal system in hES resemble or differ from phenomena observed in other pathways or other cell types. For example, different target programs are activated by sustained vs pulsatile activation of p53 and Notch, among other pathways. Also, a recent report (Frick et al., 2017 PNAS) studied the response of C2C12 cells to TGFb ligands, which belong to the Activin/Nodal family and are also transduced by Smads2/3, and showed that the pathway senses fold changes. How does this fold change detection relate to the adaptive rate-responsive behavior observed here? Discussion of these issues would help to connect this study to the broader literature and unify results from multiple studies.

We now cite and briefly discuss the work on p53 and NFkB mentioned above, however, as our focus is on adaptation in developmental signaling, we feel a detailed discussion is beyond the scope of our manuscript. Our results on ramps and pulses are not consistent with fold change detection and so we do not discuss this in the paper.

Minor Comments:In Figure 2, please include the base of the log on the y axis: is it 2, E, or something else?

It is base 2. We have now clarified this in the figure legend.

In Figure 1E there is an orange line near the y-axis. It remains unclear if this line represents the fact that after photobleaching Smad4 fluorescence does not recover to the original amplitude.

The orange line represents data from the photobleaching experiment. The drop near the y-axis is the actual bleaching event. We have now clarified this point in the figure legend.

Reviewer #3:

[…] Perhaps the biggest suggestion I can make with regards to the work is that it is a bit too dense and hard to grasp. Overall the work follows three parts: 1) An analysis of mechanism for adaptation using innovative combination of FRAP and parameter identification of ODE model. 2) analysis of gene expression dynamics response to dynamic SMAD4, and 3) implication of the differences in response dynamics on overall self-organization patterns in the in-vitro gastrulation model. I will refer to these as parts 1-3 below for brevity.I am not sure why the authors restricted themselves to only four figures, although I can make an educated guess;). To make the paper easier to follow I strongly recommend to take advantage of eLife lack of restriction on length and number of figures and use the space to elaborate much more about the work.In part 1 the author only use two paragraphs on main text to explain a lot of stuff. To understand the adaptation and the FRAP experiments one has to think about two different math models that use a parameter fitting procedure to identify underlying mechanisms. The material is mostly in the supp text, but supp text should not replace main text rather provide more details for completeness. I recommend to move some of the material from supp mat to the main text, explain the FRAP procedure, show the alternative fits for different parameters etc. This part can easily take 2-3 figures.

We thank the reviewer for these suggestions. We have now expanded the paper to 6 figures, including expanding the discussion in the FRAP section with material from the supplement.

Similarly, part II seemed rushed. The authors show how some genes follow SMAD4 dynamics and some do not. But the sparse text doesn't allow them to go into details, what is different about these genes? Is it RNA halflife? Even without additional experiments, few simple models would be very helpful to understand Figure 2A-D. I am not sure I got the point of panels 2EF, so either explain that in more details or remove those. The different gene expression response to different stimuli regimes (Figure 3) is very nice with its demonstration of the role of SMAD4 dynamics.

We have now included additional data on gene expression that rule out differences in RNA half-life (see response to reviewer 2 above). We have also expanded the discussion to better explain the gene expression data in Figures 2EF (now Figure 3EF).

Finally, the last part is very short and descriptive, The author measure different SMADs over space and time, show that there are different responses and that these change gene expression. I was left a bit confused as to why these differences arise? What aspect of self-organization is in play to cause these patterns. The fact that all these results are described in a single paragraphs of 322 words might be part of the problem.

We have now split this into two figures and expanded the explanation. We hope it is clearer now.

Minor Comments:1) The use of SMAD4 signaling in figure titles should be replaced with y-axis label saying SMAD4_{nuc}

We now use the label Smad4 (N:C) throughout to indicate that what we are measuring the ratio of nuclear to cytoplasmic Smad4.

2) The cartoon in Figure 2 is confusing, why don't all the no sequestered dots recover?

Fluorescent proteins which have been bleached never recover regardless of sequestration. In the cartoon, the non-sequestered fluorescent dots simply re-equilibrate between the nucleus and cytoplasm while the sequestered ones remain in place. We have expanded the discussion of this figure and hope it is now clearer.